# Multi-omics profiling reveals rhythmic liver function shaped by meal timing

Rongfeng Huang[1,5], Jianghui Chen[1,5], Meiyu Zhou[1,5], Haoran Xin[1,5], Sin Man Lam [2,3], Xiaoqing Jiang[1], Jie Li[1], Fang Deng[4], Guanghou Shui [2], Zhihui Zhang [1] ✉ & Min-Dian Li [1] ✉

Post-translational modifications (PTMs) couple feed-fast cycles to diurnal rhythms. However, it remains largely uncharacterized whether and how meal timing organizes diurnal rhythms beyond the transcriptome. Here, we systematically profile the daily rhythms of the proteome, four PTMs (phosphorylation, ubiquitylation, succinylation and N-glycosylation) and the lipidome in the liver from young female mice subjected to either day/sleep time-restricted feeding (DRF) or night/wake time-restricted feeding (NRF). We detect robust daily rhythms among different layers of omics with phosphorylation the most nutrient-responsive and succinylation the least. Integrative analyses reveal that clock regulation of fatty acid metabolism represents a key diurnal feature that is reset by meal timing, as indicated by the rhythmic phosphorylation of the circadian repressor PERIOD2 at Ser971 (PER2-pSer971). We confirm that PER2-pSer971 is activated by nutrient availability in vivo. Together, this dataset represents a comprehensive resource detailing the proteomic and lipidomic responses by the liver to alterations in meal timing.

Daily cycles of light and food synchronize the circadian (Latin: circa, about; diem, a day) clock system in the body[1,2]. The central circadian pacemaker is located in the suprachiasmatic nuclei (SCN) of the hypothalamus, which is tightly coupled to the light-dark cycle[3]. Extra-SCN clocks are present in almost all peripheral tissues and organs, which are synchronized to the SCN clock through neural, humoral, and metabolic signals[4,5]. The liver clock is tightly coupled to the feed-fast cycle (or meal timing)[6,7]. For example, day/sleep time-restricted feeding (DRF) reverses the liver clock within 7 days[8,9]. DRF also resets the phase of the hepatic transcriptome via the circadian clock[10–13]. Recently, we have demonstrated that the liver clock and transcriptome, but not the metabolome, entrain to DRF in female mice[14]. The discrepancy between the transcriptomic and metabolomic responses in the liver raises the question of whether alterations in the daily rhythms at the proteomic level are essential for diurnal

physiology in the liver during time-restricted feeding, at least in females[15].

Protein phosphorylation and ubiquitylation are essential PTMs in the control of the circadian clock. Mutations in *PERIOD2 (PER2)* Ser662 or its kinase genes *CSNK1D/E* (which encode CK1δ/ε) shortens the period length and decreases the stability by increased ubiquitylation[16,17]. Phosphorylation at residue Ser714 of PER1, the paralogous site of PER2 Ser662, regulates the diurnal rhythm of food intake[18]. CK1 can regulate the circadian clock and daily rhythm of energy metabolism through physical interaction with PER1/2 and likely other phosphorylation sites on PER1/2 (ref. 19). Phospho-Ser971/980/981 on PER2 have been detected in cultured cells by mass spectrometry, and shown in vitro as substrates for CK1δ[20].

It has become increasingly clear that post-translational modifications (PTMs) drive daily rhythms in the hepatic proteome.

[1]Department of Cardiovascular Medicine, Center for Circadian Metabolism and Cardiovascular Disease, Southwest Hospital, Army Medical University, Chongqing, China. [2]State Key Laboratory of Molecular Developmental Biology, Institute of Genetics and Developmental Biology, Chinese Academy of Sciences, Beijing, China. [3]LipidALL Technologies Company Limited, Changzhou, Jiangsu Province, China. [4]Department of Pathophysiology, College of High Altitude Military Medicine, Army Medical University, Chongqing 400038, China. [5]These authors contributed equally: Rongfeng Huang, Jianghui Chen, Meiyu Zhou, Haoran Xin. ✉e-mail: xyzpj@tmmu.edu.cn; mindianli@tmmu.edu.cn

Approximately 6% of hepatic proteins exhibit diurnal rhythms, which are involved in xenobiotic response, protein folding, and vesicular trafficking[21]. Protein phosphorylation in >25% of sites and >40% of proteins exhibit circadian oscillation in the liver, which includes substrates of mechanistic target of rapamycin (mTOR) kinase, insulin receptor and epidermal growth factor receptor (EGFR) signaling[22]. Protein ubiquitylation is diurnal in the liver, affecting fatty acid oxidation, glucose metabolism and peroxisome proliferator-activated receptor (PPAR) pathway signaling[23]. Protein acetylation is present mainly in mitochondrial proteins, including those related to the citric acid cycle, amino acid metabolism, and fatty acid metabolism, and this type of PTM is regulated by night/wake time-restricted feeding (NRF)[24,25]. Other PTMs, such as succinylation and N-glycosylation, regulate mitochondrial and secretory functions of the liver, respectively. However, currently there is a lack of a systems view of the regulation of proteins and PTMs during time-restricted feeding (TRF).

Hepatic lipid metabolism is regulated by the circadian clock and NRF[26–28]. Loss of circadian transcriptional repressors *Rev-Erbα/β* (*Nr1d1* and *Nr1d2*), or their corepressor *Hdac3*, results in profound diet-induced hepatic steatosis[29,30], which can be rescued by NRF[31]. Reconstitution of the core clock protein CLOCK in germline mutant mice of its gene reduces hepatic lipid accumulation[32]. Shotgun lipidomics analyses revealed that lipids in the nucleus and mitochondria of mouse livers are cycling in the morning and evening, respectively, which accounts for 30% of total lipids and is coordinated by the circadian clock and NRF[33]. We have recently developed a lipidomics technique that has an internal standard for each lipid class[34,35], which provides a quantitative and robust platform to profile the diurnal rhythms of lipids in TRF mice.

Sex differences in the robustness of circadian rhythms have been increasingly recognized. In mammals, circadian rhythms of locomotion, endocrine system, cellular metabolism, and gene expression are more sustained in females than males[36–38]. Notably, female mice exhibit a greater consolidation of diurnal rhythms of non-hepatic peripheral oscillators during DRF compared to males[7,11,14]. In the meanwhile, females are much less represented than males in circadian research[39]. Nevertheless, the liver clock and transcriptome exhibit comparable and complete entrainment to DRF from both sexes[7,11,14]. Thus, we used livers from female mice as a model to unmask the fine regulation of diurnal rhythms with respect to phase regulation.

In this study, we generated a database that profiles the proteomes of unmodified proteins and those containing four major PTMs (phosphorylation, ubiquitylation, succinylation, and N-glycosylation) in livers from TRF female mice. Samples were derived from the same cohort of mice and collected every 4 h for two complete diurnal cycles. We obtained profiles of the rhythmic proteins or modified proteins from each time-restricted feeding regimen and compared the phase regulatory pattern between DRF and NRF mice. Trans-omics integrative analyses were performed to reveal essential diurnal protein features related to meal timing. These data were matched to global lipid profiles acquired by quantitative liquid chromatography tandem mass spectrometry and diurnal transcriptome profiles. We also validated that PER2-Ser971 phosphorylation (PER2-pSer971) senses nutrient availability in hepatocytes.

## Results

### Diurnal multi-omics analysis of livers from TRF mice

To systematically explore the rhythmic liver function regulated by meal timing, we took an unbiased multi-level proteomics approach. 9-week-old female C57BL/6J mice on TRF regimens for 1 week were subjected to tissue collection at a 4-h interval over 48 h (Fig. 1a). One week is an optimal time to study molecular mechanisms while allowing the entrainment of the liver clock to be completed[8,9,14,40,41]. Mouse livers were subjected to data-independent acquisition (DIA)

shotgun proteomics. Liver protein extracts from the same cohort were separately enriched by affinity purification and analyzed by shotgun proteomics to profile protein phosphorylation, ubiquitylation, succinylation, and N-glycosylation (Fig. 1a). In addition, the liver samples from another TRF mouse cohort were dissected at a 4-h interval over one complete diurnal cycle and subjected to quantitative lipidomics analysis (Fig. 1a).

The multi-omics study robustly identified tens of thousands of PTM sites and proteins. These included 10,874 proteins, 30,463 phospho-sites, 11,136 ubiquityl-sites, 7944 succinyl-sites, and 5245 N-glycosyl-sites, of which 17,359 phospho-sites, 8430 ubiquityl-sites, 7446 succinyl-sites, and 4305 glycosyl-sites are quantified (<30% missing values across samples) (Fig. 1b). In addition, we quantified 571 lipid species covering 29 lipid classes in the lipidomics study, and 50 lipid species covering 4 lipid classes that belong to N-Acyl-Phosphatidylethanolamine (NAPE), endocannabinoids (EC) and acyl amino acids (acyl-AA) (Fig. 1b).

We sought to identify PTM sites on the circadian clock proteins. In total, 57 sites on circadian clock proteins that can be modified by phosphorylation or ubiquitylation were identified in this multi-omics study (Supplementary Table 1), including the recently reported mouse PER2 Ser693/Ser697 and Cryptochrome 1 (CRY1) Ser588 sites[22]. For example, Rev-Erbα/NR1D1 lysine 456 was identified as a ubiquitylation site (Supplementary Fig. 1a). Lysine 456 mediates the physical interaction between NR1D1 and PER2, the loss of which reduces the interaction by 60% (ref. [42]). Rev-Erbα/NR1D1 Ser279 was identified as a potential phosphorylation site among three candidate sites (Supplementary Fig. 1b), the mutation of which reduces the interaction with the FBXW7 E3 ubiquitin ligase and subsequent degradation[43]. Notably, PER2-pSer971 had the most peptide spectrum matches (Supplementary Fig. 1c), but the in vivo regulation of this modification has not been elucidated.

To determine daily rhythms of proteins and lipids in response to TRF, we applied a stringent algorithm. The multi-omics profiles were subjected to three different rhythmicity detection methods, including a nonparametric detection method (Rhythmicity Analysis Incorporating Nonparametric method, or RAIN), a parametric method (CircaCompare), and a hybrid method (MetaCycle)[44–46]. Diurnal rhythmicity is determined by adjusted $P$-values < 0.05 in RAIN and MetaCycle and a $P$-value < 0.05 in Circacompare. Consequently, we detected robust diurnal rhythms in 45% (2139/4778) of phospho-proteins, 38% (483/1277) of ubiquityl-proteins, 34% (1983/5773) of proteins, 16% (196/1252) of N-glycosyl proteins, and 1.9% (24/1286) of succinyl-proteins in the mouse livers (Fig. 1b). The phospho-proteome appears much more rhythmic than the other forms of PTMs, which is consistent with the high degree of circadian rhythms of the phospho-proteome in the liver of ad libitum-fed mice[22]. In contrast, protein succinylation exhibits the lowest degree of diurnal rhythm, at least within the depth of the current proteome profiling.

### Entrainment of the diurnal phospho-proteome by DRF

Considering the great percentage of diurnal proteins in the phospho-proteome, we examined diurnal rhythms in the mouse phospho-proteome first. Most diurnal phospho-proteins oscillated around ZT4 under DRF and at ZT8 and ZT20 under NRF (Supplementary Fig. 2a). The amplitude and relative amplitude (the ratio between amplitude and baseline) of diurnal phospho-proteins were significantly lower under DRF compared to NRF (Supplementary Fig. 2a). Pathway enrichment based on the phase of diurnal phospho-proteins revealed that almost all rhythmic pathways peak in the sleep phase (ZT0-8) during DRF and concentrate in the first half of the daytime (ZT3-6) (Supplementary Fig. 2b). This is represented by the regulation of protein targeting (ZT3), cell matrix adhesion (ZT4), response to calcium ion (ZT5) and macroautophagy (ZT6) (Supplementary Fig. 2c). In the NRF group, enriched pathways almost exclusively peaked in the

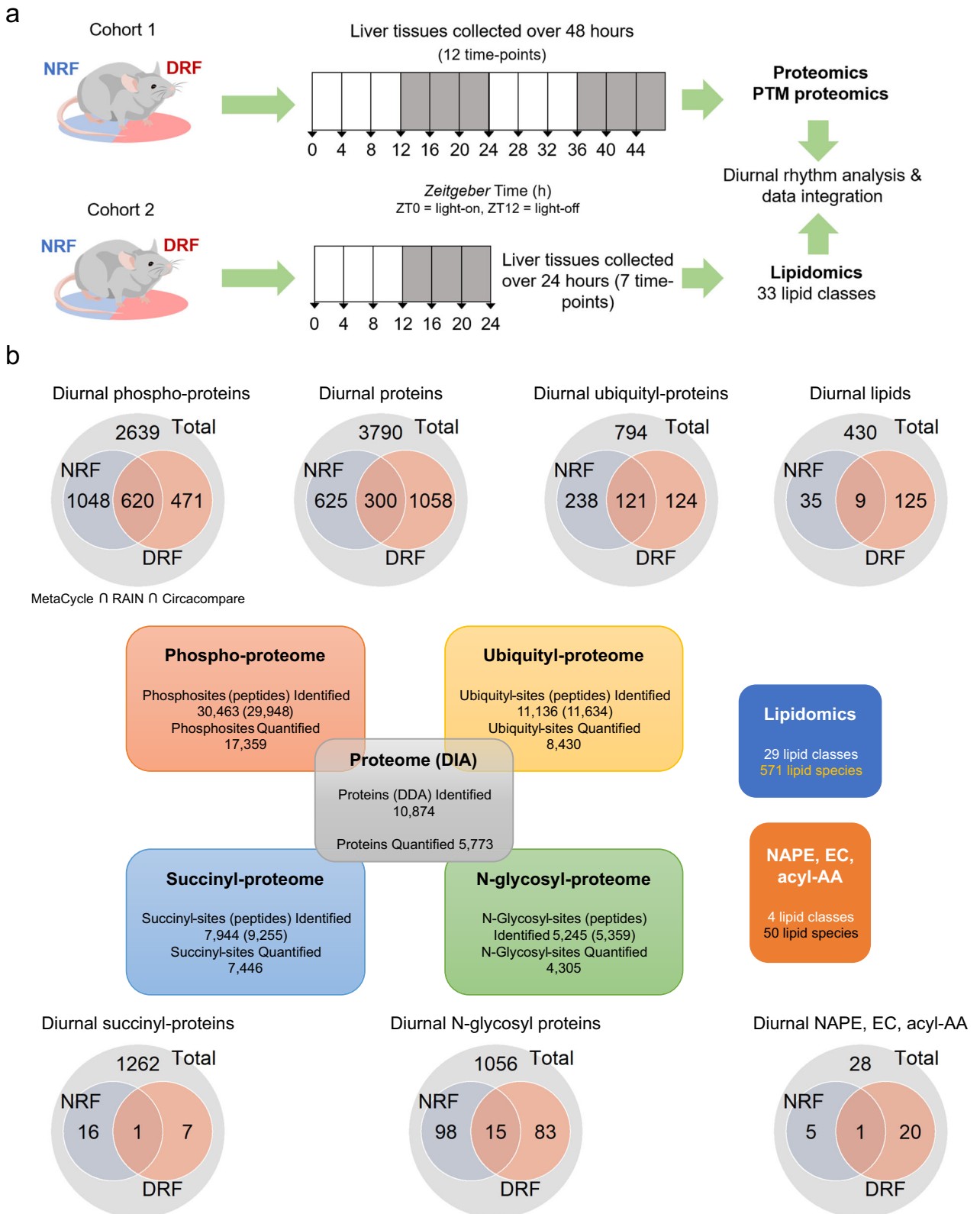

**Fig. 1 | Diurnal multi-omics analysis of the mouse liver under time-restricted feeding. a** A schematic diagram illustrating the workflow of the multi-omics study on the effects of meal timing in the liver from 9-week-old C57BL/6J female mice. The intervention of meal timing lasts for 1 week. Samples were dissected every 4 h for 2 days (n = 48 mice per group). Proteomics, phosphoproteomics and ubiquityl-proteomics (n = 4 mice per time-point), succinyl-proteomics and N-glycosyl-proteomics (n = 2 mice per time-point) were performed and integrated. Data were matched with diurnal lipidomics (n = 4 mice per time point for 7 time points within one diurnal cycle). **b** Interaction of diurnal molecules in each omics dataset, and summary of identified and quantified molecules in the multi-omics data. Numbers in blue and red circles of Venn diagrams denote the number of rhythmic molecules, while numbers in the gray area denote arrhythmic molecules. Circadian rhythmicity was determined by the algorithms MetaCycle (adjusted $P$-value < 0.05), RAIN (adjusted $P$-value < 0.05), and CircaCompare ($P$-value < 0.05). DRF, day/sleep time-restricted feeding; NRF, night/wake time-restricted feeding; NAPE, N-acylphosphatidyl ethanolamine; EC, endocannabinoid; acyl-AA, acyl amino acids. Source data are provided as a Source data file.

second half of the sleep phase (ZT6-11) (Supplementary Fig. 2b) and included lymphocyte homeostasis (ZT7), regulation of cell matrix adhesion (ZT8), mitochondrial membrane organization (ZT9) and drug catabolic process (ZT10) (Supplementary Fig. 2d).

Next, we examined the interaction between DRF and NRF in the diurnal phospho-proteome. 620 of the 4778 diurnal phospho-proteins were robustly cycling under both DRF and NRF, which were dubbed as dual-cycling phospho-proteins (Fig. 2a). The phase distribution of these dual-cycling phospho-proteins was bimodal. Dual-cycling phospho-proteins from DRF mice peaked at ZT4 and ZT12, whereas their counterparts from NRF mice peaked at ZT6 and ZT20 (Fig. 2b). 136 dual-cycling phospho-proteins from DRF mice remain phase-locked to their peak time during NRF, and 143 (or 23.1% of 620) were inverted in phase by DRF (Fig. 2c). This is unexpected, especially considering that 60% of dual-cycling transcripts are inverted in phase by DRF[14].

We performed pathway analysis based on phase-shifts and found that most rhythmic pathways were shifted in phase for 4–5 h by DRF, compared to NRF (Fig. 2d). Rhythmic pathways that were poorly entrained by meal time included inositol lipid-mediated signaling (phase-shift 3 h, Kuiper test, $q$-value = 0) and receptor-mediated endocytosis (phase-shift 4 h, Kuiper test, $q$-value = 0) (Fig. 2e). We found that the receptor-ligand machinery involved in lipid clearance exhibited robust diurnal rhythms under both DRF and NRF, but it was not altered in phase between DRF and NRF, as indicated by diurnal phosphorylation profiles of LRP1 (LDL receptor-related protein 1), LDLRAP1 (LDL receptor adaptor protein 1) and LRPAP1 (LRP associated protein 1) (Fig. 2f).

The most food-entrainable pathways exhibited a phase-shift of 6–9 h and included monocarboxylic acid biosynthetic process (phase-shift 9 h), regulation of circadian rhythm, fatty acid metabolism, mitochondrial matrix and transcriptional corepressor activity (Fig. 2g).

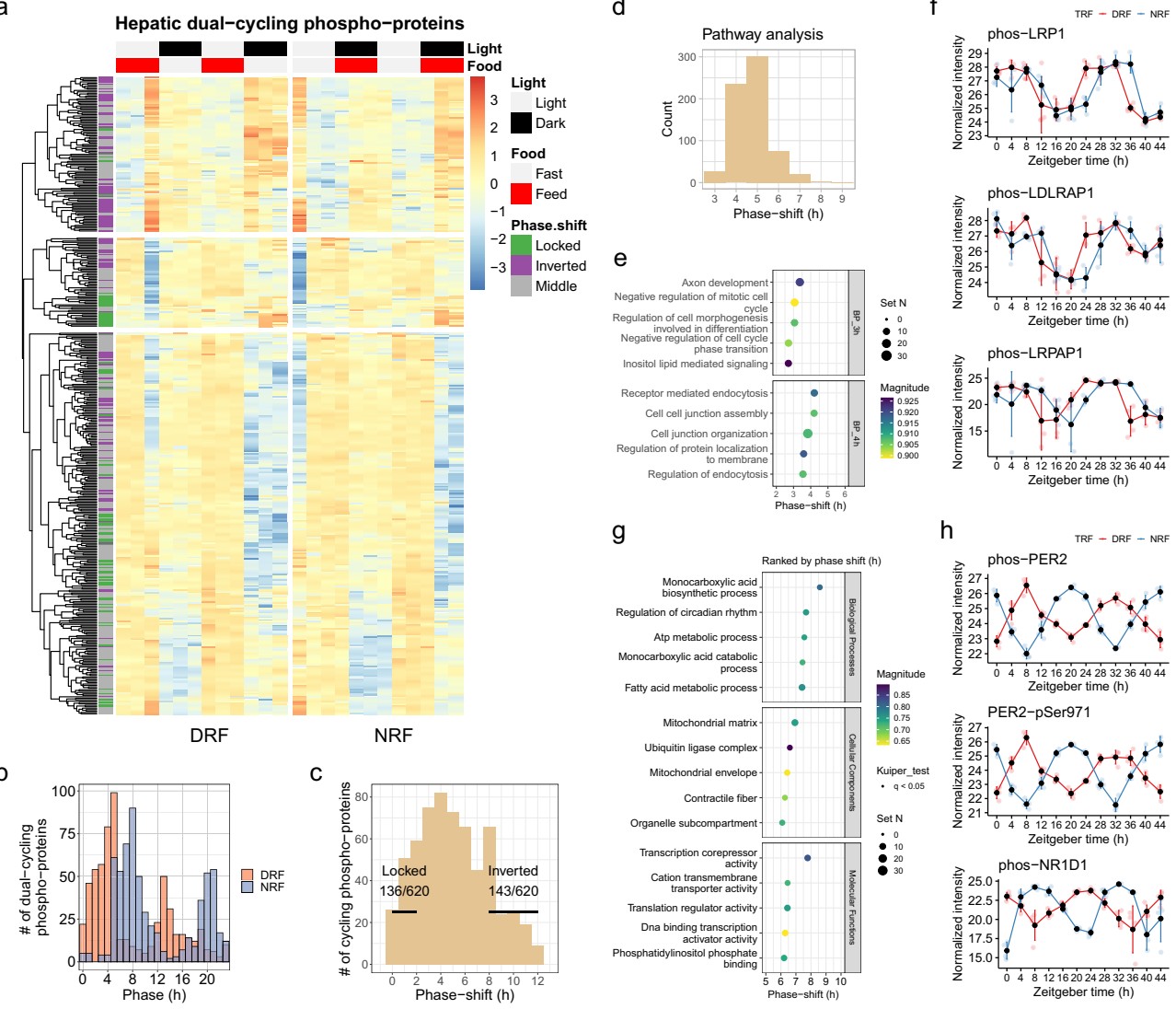

**Fig. 2 | Entrainment of the diurnal phosphoproteome by DRF. a** Expression profiles of phosphorylated proteins in the mouse liver that are diurnal under both DRF and NRF (referred to as dual-cycling phospho-proteins). Average expression levels were log2-transformed and scaled ($n$ = 48 mice per group). **b, c** Histogram showing the distribution of phase (**b**) or phase-shift (**c**) of dual-cycling phospho-proteins in the mouse liver. Locked, phase shift [0, 2 h]; inverted, phase shift [8, 12 h]; numbers denote phase-locked/inverted rhythmic phosphoproteins over total dual-cycling phosphoproteins. **d** Histogram showing the phase distribution of enriched GO: Biological Process pathways, as measured by phase set enrichment analysis (PSEA, Kuiper test $q$ < 0.05) of hepatic dual-cycling phospho-proteins. **e, f** Display of the least (**e**) or the most (**f**) phase-shifted pathways. Pathway terms are ranked based on the average phase shift and $q$-values, and the top 5 are displayed. **g, h** Diurnal profiles of representative phospho-proteins involved in the receptor-mediated endocytosis (**g**) and regulation of circadian rhythm (**h**) in the mouse liver. Data are presented as mean values ± standard deviation, $n$ = 4 mice per time point for 12 time points. Source data are provided as a Source data file.

Notably, phosphorylated forms of the core clock proteins, such as PER2 and Rev-Erbα/NR1D1, were inverted in phase by DRF compared to NRF (Fig. 2h). PER2-pSer971 exhibited robust diurnal rhythms in the livers of both DRF and NRF mice, and its diurnal abundance was completely reversed in phase by DRF compared to NRF. Overall, 23.1% (143/620) of the dual-cycling phospho-proteins were entrained by DRF in mouse livers compared to NRF, despite a robust complete entrainment of the circadian clock by meal timing.

### Features associated with meal timing in the diurnal proteome

Next, we explored diurnal rhythms of the hepatic proteome. Proteomics by a data-independent acquisition mode provided us with a deep, robust, and quantitative approach to analyze diurnal rhythms among 5773 hepatic proteins. Most diurnal proteins peaked around ZT20 and ZT0 during DRF and NRF, respectively (Supplementary Fig. 3a). The relative amplitude is slightly smaller under DRF compared to NRF (Wilcoxon test $P = 0.026$), but the amplitude is not statistically different between DRF and NRF (Supplementary Fig. 3a). Notably, 28.0% or 380 of the 1358 diurnal proteins from DRF mice peaked at ZT20. In line with this, enriched DRF diurnal pathways peaked around ZT20 (Supplementary Fig. 3b), including xenobiotic metabolic process (ZT18), serine family amino acid metabolism (ZT19), cellular hormone metabolic process (ZT19), neurotransmitter metabolic process (ZT20) and monosaccharide biosynthetic/catabolic process (ZT21) (Supplementary Fig. 3c). Enriched NRF diurnal pathways peaked from ZT23 to ZT1 (Supplementary Fig. 3b), including xenobiotic metabolic process (ZT22), regulation of cholesterol biosynthetic process (ZT23), Protein N-linked glycosylation (ZT23), lipid storage (ZT24 or ZT0) and peroxisome organization (ZT1) (Supplementary Fig. 3d). These features seen in livers from NRF mice are found in ad libitum-fed mouse livers collected in constant darkness, as well[21].

As shown in Fig. 1b, 300 diurnal proteins are shared between DRF and NRF groups and exhibit robust diurnal rhythms in mouse livers (Fig. 3a). The phase distribution of dual-cycling proteins peaks around ZT20 and ZT1 under DRF and NRF, respectively (Fig. 3b). 81 (27%) of these proteins from DRF mice remained phase-locked to their peak time during NRF, whereas 52 (17.3%) were inverted in phase by DRF (Fig. 3c). Pathway enrichment analysis based on phase-shifts revealed that most rhythmic pathways were shifted in phase for 4–5 h (Fig. 3d), and the most enriched pathways, e.g., triglyceride/cholesterol metabolic process, regulation of peptide secretion and microbody, exhibited not more than 8 h of phase-shift (Fig. 3e).

We examined diurnal rhythmicity of representative proteins in the lipid metabolic pathway (Circacompare method, $P < 0.05$), and found that the rhythmicity of the lipid droplet coat protein Perilipin-2 (PLIN2) was shifted in phase for 7.5 h and increased significantly in baseline (mesor) by DRF compared to NRF (Fig. 3f). DRF abolished the rhythmicity of PLIN3 protein, whereas it shifted the phase of PLIN5 protein by 6.8 h without altering the baseline (mesor) and amplitude (Fig. 3f). Proteins involved in fatty acid desaturation (FADS2), very-long/long chain fatty acyl coenzyme A biosynthesis (SLC27A2, ACSL4) and oxidation (ACOX2) and degradation of bioactive lipids (FAAH) were all increased in baseline abundance by DRF in the liver and shifted in phase for various hours (Fig. 3f). Overall, lipid metabolic proteins exhibit a pathway-level increase in abundance and a phase-shift of 8 h.

### Entrainment of the diurnal ubiquityl-proteome by DRF

Diurnal phase analysis of the di-glycine peptide-enriched ubiquityl-proteome profiles performed in our study revealed that diurnal ubiquityl-proteins from DRF mice were evenly distributed throughout the diurnal cycle, whereas diurnal ubiquityl-proteins from NRF were mainly in the dark/active phase in mouse livers (Supplementary Fig. 4a). Both the average relative amplitude and the average amplitude were decreased by DRF compared to NRF (Supplementary Fig. 4a). The 26S proteasome activity was diurnal in livers from DRF mice and peaked around ZT20.5 (ZT19.7 from Circacompare method, adjusted $P = 0.0076$; ZT21 from RAIN $P = 0.022$); however, it was not rhythmic in livers from NRF mice (Supplementary Fig. 4b). This is associated with a significant decrease in the baseline (mesor) activity.

Pathway enrichment analysis revealed that rhythmic pathways under DRF reached peak time hours from ZT10 to ZT20, spanning from the late feeding/day phase to the fasting/night phase (Supplementary Fig. 4c). Diurnal signatures found in the hepatic ubiquityl-proteome from DRF mice are represented by lipid/steroid metabolic process (ZT10), xenobiotic metabolic process (ZT11), carbohydrate metabolic process (ZT13) and response to lipid (ZT14) (Supplementary Fig. 4d). In contrast, a diverse array of pathways is rhythmic and peak around ZT17 in the ubiquityl-proteome from NRF mice (Supplementary Fig. 4c), including alpha/cellular amino acid metabolic process (ZT16), lipid homeostasis (ZT17), lipid modification (ZT18) and response to metal ion (ZT19) (Supplementary Fig. 4d).

Next, we examined the interaction between DRF and NRF in the diurnal ubiquityl-proteome. 121 dual-cycling ubiquityl-proteins oscillated robustly in the liver (Fig. 4a). 35 of these proteins from DRF mice remained phase-locked to the original peak time during NRF, and 25 rhythmic ubiquityl-proteins were inverted in phase by DRF compared to NRF (Fig. 4b). Pathway enrichment analysis identified 92 rhythmic pathways among dual-cycling ubiquityl-proteins, most of which were shifted in phase for 4 h between DRF and NRF (Fig. 4c). Lipid metabolic process (phase-shift 6 h) is a signature diurnal pathway that entrained to feeding rhythm, whereas protein localization to endoplasmic reticulum (phase-shift 3 h) was among the poorly food-entrainable pathways, as indicated by diurnal ubiquitylation patterns (Fig. 4d). In line with these findings, the rhythmic ubiquitylation (ub) pattern of very long-chain acyl-CoA synthetase (SLC27A2) was inverted in phase by DRF compared to NRF (Circacompare method, phase-shift: 8.8 h, $P = 7.44E-15$), which could be attributed to diurnal ubiquitylation of ub-K398 and ub-K419 of the protein (Circacompare method, phase difference: 8.6 h, $P = 3.21E-05$ and 8.4 h, $P = 7.73E-10$, respectively) (Fig. 4e). Thus, lipid metabolic process is a stellar feature of the diurnal ubiquityl-proteome that is responsive to meal timing.

### Clock regulation of fatty acid metabolism is a key feature of the hepatic diurnal proteome under DRF

To identify diurnal features driven by DRF, we took a supervised N-integration sparse discriminant analysis and integrated diurnal multi-omics data from the same mouse liver samples. Proteins, phospho-proteins, and ubiquityl-proteins with daily rhythms in at least one TRF condition served as the input data to mixOmics analysis[47,48]. The modeling applies the N-integrative supervised analysis with DIABLO, because the same N samples are measured on different 'omics platforms. We divided the input data into three subsets based on the time of day; i.e., morning subset (sampled at ZT0 and ZT4, $n = 16$ per group), evening subset (sampled at ZT8 and ZT12, $n = 16$ per group) and night subset (sampled at ZT16 and ZT20, $n = 16$ per group), and fed into the statistical modeling. The results showed that the evening subset returned the best-fit model with an overall balanced error rate (BER) of 0.0312 compared to the other two subsets (morning subset BER = 0.278 and night subset BER = 0.203).

Next, we focused on the mixOmics model based on the evening subset of multi-omics data and found that the statistical model could clearly segregate TRF groups in the phospho-proteome and proteome dataset, and to a lesser degree in the ubiquityl-proteome (Fig. 4f). Importance plotting shows the rank and weight of the diurnal proteins that distinguish DRF from NRF. We found that phosphorylated heme-binding protein 1 (p-HEBP1) and phosphorylated Rev-Erbα/NR1D1 (p-NR1D1), which is a nuclear receptor for heme[49], and p-PER2 contributed the most (Fig. 4g).

A list of 20 diurnal proteins from the proteomics dataset contributes to the ability to distinguish the effects of DRF from NRF on

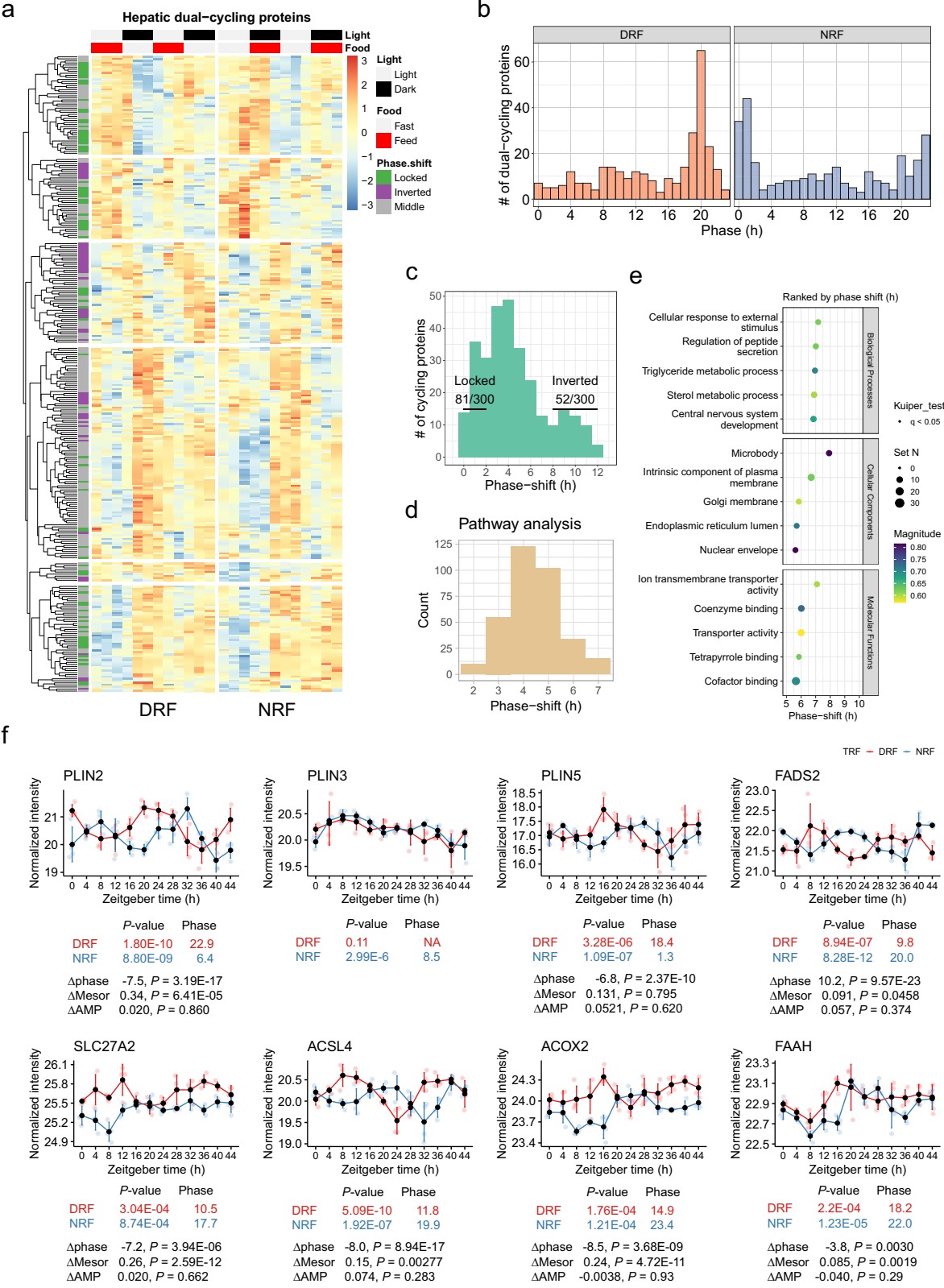

**Fig. 3 | Features associated with meal timing in the diurnal proteome.**
**a** Expression profiles of proteins in the mouse liver that are rhythmic under both DRF and NRF (referred to as dual-cycling proteins). Average expression levels were log2-transformed and scaled ($n = 4$ mice per time point for 12 time points). **b, c** Distribution of phase (**b**) and phase-shift (**c**) for dual-cycling hepatic proteins. Locked, phase shift [0, 2 h]; inverted, phase shift [8, 12 h]; numbers denote phase-locked/inverted rhythmic proteins over total dual-cycling proteins. **d** Histogram showing the phase distribution of enriched GO: Biological Process pathways, as measured by phase set

enrichment analysis (PSEA, Kuiper test $q < 0.05$) of hepatic dual-cycling proteins. **e** Display of the most phase-shifted pathways. Pathway terms are ranked based on the average phase shift and $q$-values, and the top 5 are displayed. **f** Diurnal expression of proteins involved in lipid storage and fatty acid metabolism in the mouse liver (Circacompare method, $P < 0.05$, two-sided). Data are presented as mean values ± standard deviation, $n = 4$ mice per time point for 12 time points. Source data are provided as a Source data file.

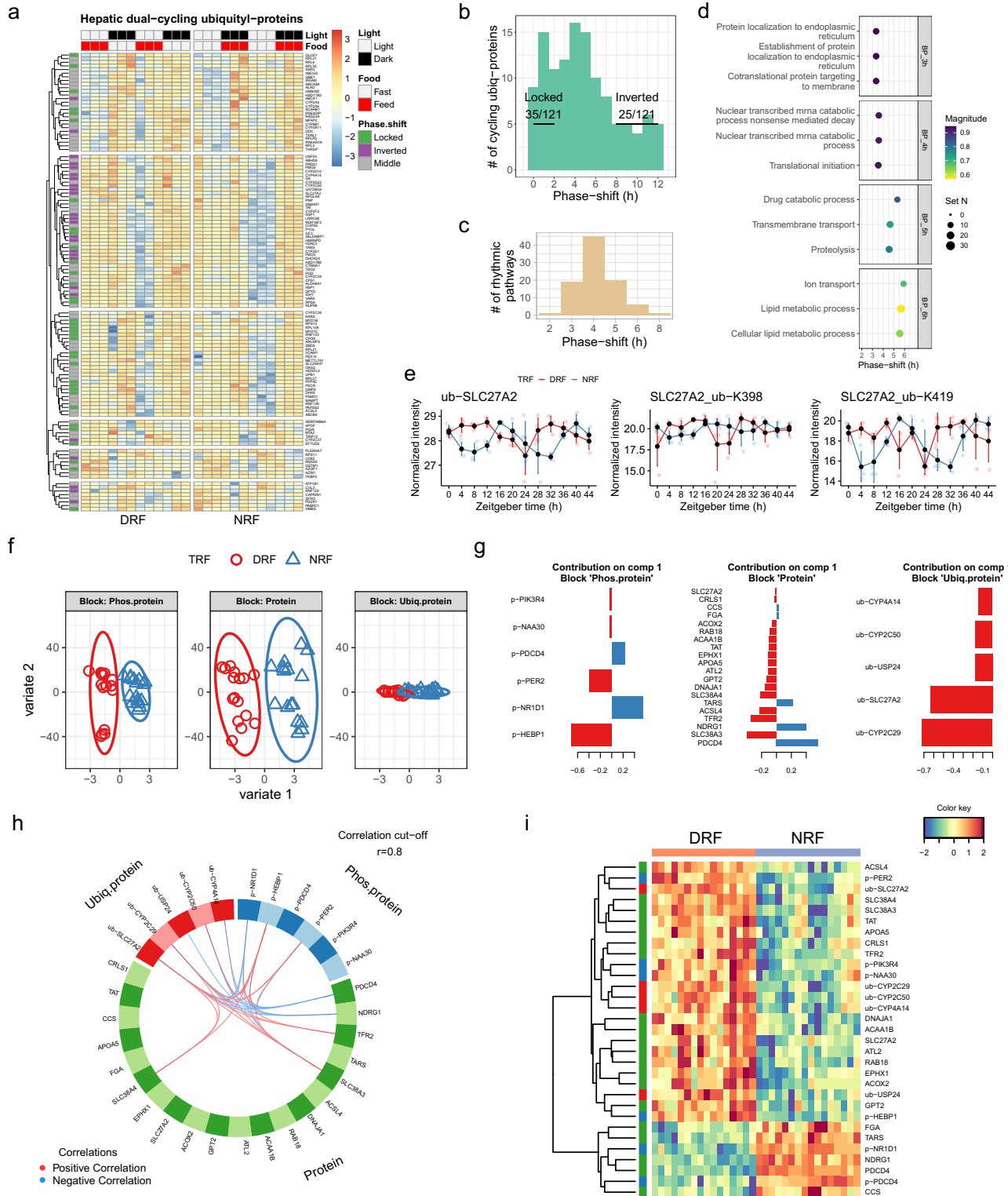

the liver diurnal rhythm (Fig. 4g). Enzymes involved in fatty acid metabolism are also represented, such as ACSL4, ACOX2, and SLC27A2. Strikingly, four of the five contributing proteins in the diurnal ubiquityl-proteome (CYP2C29, SLC27A2, CYP2C50, and CYP4A14) are involved in arachidonic acid metabolism, which is part of fatty acid metabolism (Fig. 4g). To reveal the correlation among the contributing features, we generated a circular correlation plot and found that phospho-proteins and ubiquityl-proteins are tightly correlated (Fig. 4h). In particular, p-PER2 is positively correlated with

ubiquityl (ub)-SLC27A2 whereas the latter is negatively correlated to p-NR1D1.

We further confirmed this feature by examining the clustered image map, where p-PER2 and many of the fatty acid metabolic enzymes, e.g., ACSL4, SLC27A2 and its ubiquityl form, ubiquityl forms of CYP2C29, CYP2C50, and CYP4A14, show greater weight in DRF compared to NRF (Fig. 4i). Overall, our integrative model indicates that the circadian clock, particularly represented by p-PER2, and its tightly associated proteins in the fatty acid

**Fig. 4 | Clock regulation of fatty acid metabolism is a key feature of the hepatic proteome under DRF. a** Expression profiles of dual-cycling ubiquityl-proteins in the mouse liver. Average expression levels were log2-transformed and scaled (*n* = 48 mice per group). **b** Histogram showing the phase-shift distribution of dual-cycling hepatic ubiquityl-proteins. Locked, phase shift [0, 2 h]; inverted, phase shift [8, 12 h]; numbers denote phase-locked/inverted rhythmic ubiquityl-proteins over total dual-cycling ubiquityl-proteins. **c, d** Histogram showing the phase-shift distribution (**c**) and representative pathways (**d**) of enriched GO: Biological Process pathways, as measured by phase set enrichment analysis (PSEA, Kuiper test *q* < 0.05) of hepatic dual-cycling ubiquityl-proteins. **e** Diurnal expression of ubiquitylated (ub-) SLC27A2 in mouse liver. Data are presented as mean values ± standard deviation, *n* = 4 mice per time point for 12 time points. **f–i** Sample plot per Omics (**f**), loading plot displaying the contribution (loading weight) of each feature selected from the first component per Omics in an increasing order of importance from the bottom up (**g**), circos plot showing the positive (negative) correlation (r > 0.8) between selected features (rhythmic proteins) as indicated by the red (blue) links (**h**) and clustered image maps of component 1 features (**i**) from the N-integrative supervised analysis with multi-omics data sampled in the evening (ZT8 and ZT12, *n* = 16 per group). Diurnal proteins are included for this analysis. Phos.protein, phosphorylated proteins; Ubiq.protein, ubiquitylated proteins. Source data are provided as a Source data file.

metabolism pathway, is a key diurnal feature in response to TRF in the mouse liver.

## Diurnal analyses of the N-glycosyl- and succinyl-proteomes

We next examined the diurnal rhythms of N-glycosyl-proteins and succinyl-proteins, the roles of which are unknown in circadian rhythm. Most of the rhythmic N-glycosyl proteins peaked from ZT12 to ZT18 during DRF or peaked around ZT16 during NRF (Supplementary Fig. 5a). Both the average relative amplitude and average amplitude were reduced by DRF compared to NRF (Supplementary Fig. 5b). In line with the phase distribution of individual N-glycosyl proteins, rhythmic pathways from either DRF or NRF mice were phasic in the early physically active phase (ZT13-16 or ZT14-17) (Fig. 5a). Rhythmic pathways found in DRF mouse livers include cell projection organization (ZT13), endocytosis (ZT14), protein catabolic process (ZT15) and activation of immune response (ZT16) (Fig. 5b). Rhythmic pathways from NRF mice are represented by transmembrane ion transport (ZT15-16) (Fig. 5c). Among the 15 dual-cycling N-glycosylated proteins (Fig. 5d), five were reversed in phase by DRF compared to NRF (Fig. 5e). For example, apolipoprotein B (APOB) is among the five phase-inverted glycosyl-proteins, with proteins governing lipid uptake, e.g., angiopoietin-like 3 (ANGPTL3), exhibiting a moderate phase shift between DRF and NRF (Fig. 5f).

Considering that a rather small percentage of succinyl-proteins cycled in the livers of TRF mice (Fig. 1b), we examined diurnal rhythms among individual sites of succinylation. We found that diurnal succinyl-sites are clustered into two groups that peak around ZT10 and ZT20, respectively, under DRF, whereas succinyl-sites are generally scattered around the clock except in one peak at ZT16 under NRF (Supplementary Fig. 5c). There is no significant difference between DRF and NRF, with respect to the average amplitude (Supplementary Fig. 5d). In total, there are 97 diurnal succinyl-sites, seven sites of which are shared between DRF and NRF mice (Fig. 5g). Consistent with the fact that protein succinylation occurs mainly in mitochondria[50], almost all of the dual-cycling succinyl-sites were found in mitochondrial proteins, except K414 of CYP2D26 (Fig. 5h). Rhythmic succinylation at K644 of mitochondrial trifunctional enzyme alpha subunit HADHA and K93 of SUCLA2 were inverted in phase by DRF compared to NRF, and these are the only two food-entrainable succinyl-sites identified in this study (Fig. 5h, i).

## Diurnal lipidomics profiling corroborates the entrainment of fatty acid metabolism by DRF

Diurnal and integrative analyses of the multi-level proteomics revealed that diurnal regulation of fatty acid metabolism in the mouse liver is an essential feature associated with meal timing. To validate this idea, we analyzed the lipidome profiles in a new cohort of mice subjected to different TRF conditions. As shown in Fig. 1b, DRF entrains robust diurnal rhythms among 155 lipids covering 33 lipid classes in the liver, which is roughly three times more than the number of lipids in livers from NRF mice. A vast majority of diurnal lipids from DRF mice peaked at ZT1, followed by a second peak at ZT12 (Fig. 6a). Diurnal lipids from NRF mice peaked from ZT16 to ZT20 (Fig. 6a). Strikingly, 7 of the 10

dual-cycling lipids from DRF mice were phase-inverted, compared to NRF (Fig. 6a).

The phase-inverted lipids consist mainly of Bis-monoacylglycerophosphate (BMP) and glycerophospholipids (Fig. 6b). BMP is primarily present in endosomal and lysosomal compartments[51]. BMP species peaked in the sleep and wake time under DRF and NRF, respectively, in synchrony with the feeding time (Fig. 6c). Acylcarnitine species were either moderately phase-shifted or phase-locked in the liver from DRF mice compared to NRF (Fig. 6b, c). Remarkably, DRF induced robust diurnal rhythms among more than 80 species of triacylglycerols (TAGs) and 8 species of diacylglycerols (DAGs) in the liver (Fig. 6d and Supplementary Fig. 6). In contrast, NRF entrained diurnal rhythms of one TAG species in the liver, while it entrained robust diurnal rhythms of acylcarnitines - essential for fatty acid oxidation - and phosphoglycerolipids, primarily phosphatidylserines (PSs) (Fig. 6d).

We matched the profiles of TAGs and DAGs with those of monoacylglycerols (MAGs) and biophysically relevant lipids by targeted quantification profiling. Diurnal rhythms in *N*-acylphosphatidyl ethanolamines (NAPEs) and C20:1 MAG (MAG 20:1) reached the peak time in the sleep phase during DRF (Fig. 6e). While arrhythmic during DRF, myristoleoyl MAG (MAG14:1), oleoyl MAG (MAG 18:1) and *N*-acylethanolamines (NEA, endocannabinoids) as a class (represented by oleoyl ethanolamine/18:1-EA and palmitoyl ethanolamine/16:0-EA) exhibited robust daily oscillation in livers from NRF mice (Fig. 6e). This rhythmic pattern of precursors and components of endocannabinoids coexists with the rhythmicity of the fatty acid amide hydrolase FAAH (Fig. 3d).

## Integrative analysis of diurnal transcriptome and multi-level proteomes

Next, we compared this diurnal multi-level proteomics dataset with a recently published diurnal transcriptomics dataset (GSE150380)[14] and found that rhythmic proteins (including unmodified and one of the four PTMs) and rhythmic mRNAs had 489 matched pairs based on the source gene (Fig. 7a). A comparison among rhythmic unmodified proteins, rhythmic PTM proteins, and rhythmic mRNAs revealed that 77 genes exhibited diurnal rhythms in all three groups and 947 between two groups (Fig. 7a). In contrast, 2,294 rhythmic mRNAs did not have matched diurnal rhythmic proteins (Fig. 7a).

Among the 489 mRNA-protein pairs, there is one gene (the gluconeogenic gene *Pck1*) that exhibited diurnal rhythmicity at four and the most levels of omics, followed by five genes (*Plin5, Got1, Sun2, Hmgcr,* and *Abcg5*) with diurnal activity across three levels of omics (Fig. 7b). Phase plot showed that almost all these genes exhibited various degrees of phase delay from a transcript-level rhythm to protein-level rhythms (Fig. 7b). This is matched with the patterns seen in circadian clock genes, such as *Per2* and *Nr1d1*. Remarkably, the succinylated PCK1 rhythm exhibited an 8-h phase delay compared to its mRNA rhythm under NRF (Fig. 7b, c). The glutamic-oxaloacetic transaminase gene *Got1* is an exception in that glycosylated GOT1 rhythm was 10.7 h advanced in phase compared to either protein or mRNA rhythms under DRF (Fig. 7b, c).

We analyzed the mRNA-protein pairs and found that the connections between diurnal multi-level proteomes and transcriptome

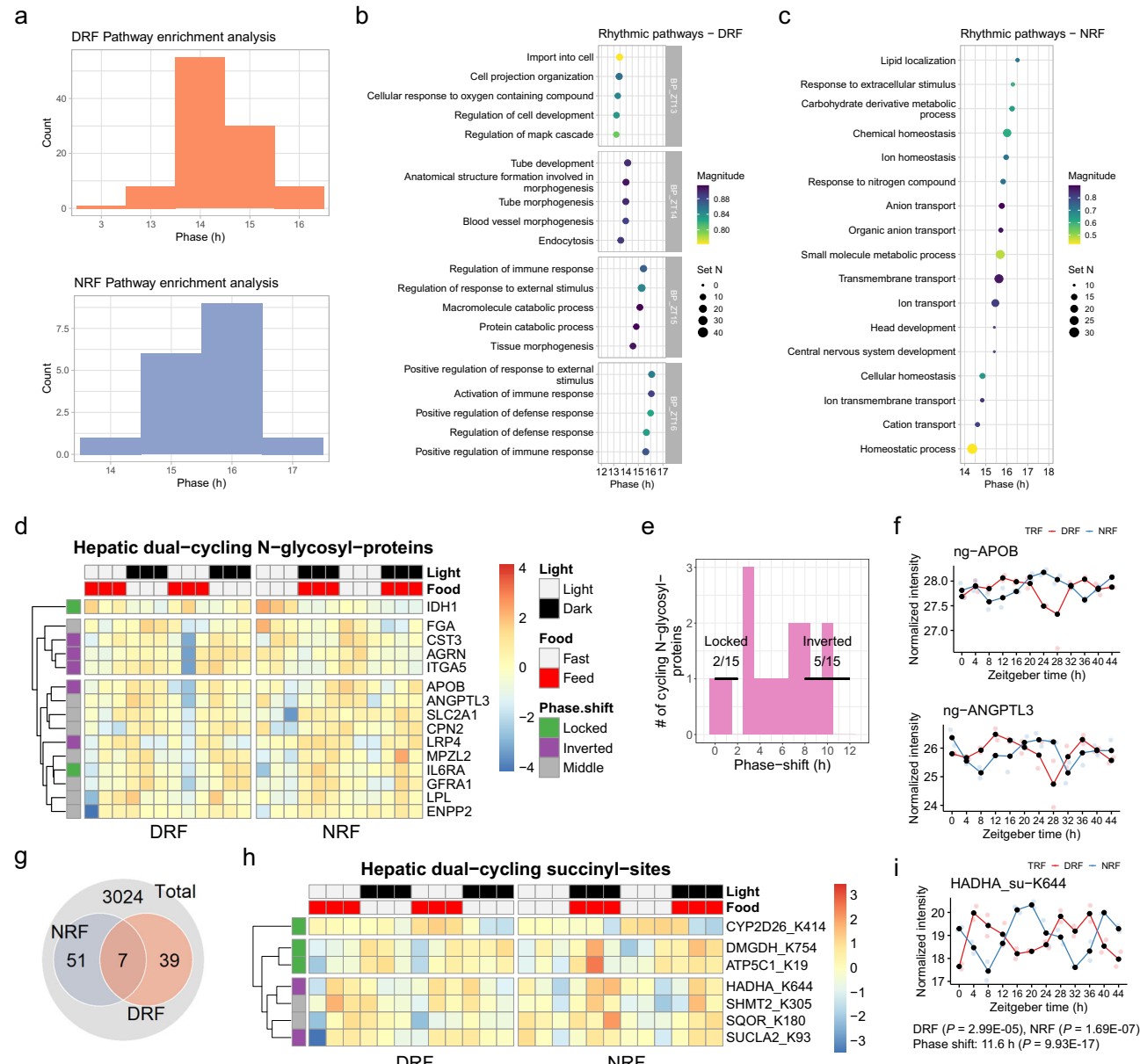

**Fig. 5 | Integrated analyses of diurnal transcriptome and multi-level proteomes. a** Histogram showing the phase distribution of enriched rhythmic pathways under DRF or NRF, as measured by PSEA of cycling proteins in livers from DRF or NRF female mice (Kuiper test, $q < 0.05$). **b, c** Representative rhythmic pathways in mouse livers under DRF (**b**) or NRF (**c**), as shown by the hour of their estimated peak time. **d** Diurnal profiles of dual-cycling N-glycosyl proteins in the mouse liver. Average expression levels were log2-transformed and scaled ($n = 24$ mice per group). **e** Histogram showing the phase-shift distribution of dual-cycling hepatic N-glycosyl proteins. Locked, phase shift [0, 2 h]; inverted, phase shift [8, 12 h];

numbers denote phase-locked/inverted rhythmic N-glycosyl proteins over total dual-cycling N-glycosyl proteins. **f** Diurnal profiles of N-glycosylated (ng-) APOB and ANGPTL3 in mouse liver, as measured by affinity purification-mass spectrometry. $N = 24$ mice per group. **g** Interaction of diurnal succinyl-sites between DRF and NRF. **h** Diurnal profiles of dual-cycling succinyl-sites in the mouse liver. Average expression levels were log2-transformed and scaled ($n = 24$ mice per group). **i** Diurnal profile of succinylation at K644 of HADHA (HADHA_su-K644). Circa-Compare method ($n = 24$ mice per group, $P < 0.05$, two-sided test). Source data are provided as a Source data file.

exhibited a similar pattern between DRF and NRF (Fig. 7d). Transcript rhythms were mainly matched with rhythms in the unmodified protein or phosphorylated proteins, whereas unmodified protein rhythms were mainly matched with rhythms in mRNA, phosphorylated or ubiquitylated proteins. These findings revealed the complex network with regards to the biogenesis of diurnal rhythms in the liver.

Next, we compared the rhythmic pathways among different levels of omics, as measured by PSEA of rhythmic proteins and genes (Kuiper test, $q < 0.05$). Like with the gene-level comparison, most pathways had no matched rhythms across the omics under either DRF or NRF (Supplementary Fig. 7a). Connectivity maps for the rhythmic pathways

(224 and 185 for DRF and NRF, respectively) revealed that strong connections under NRF including the mRNA-phosphorylation and phosphorylation-ubiquitylation links were replaced by connections such as the unmodified protein-glycosylation and phosphorylation-glycosylation links under DRF (Fig. 7e). Comparative analysis of rhythmic pathways between transcriptome and proteomes showed that non-coding RNA processing and metabolic process (ZT8-10) were found at the transcript and phosphorylated protein levels under NRF, while pathways related to translation (ZT18-20) were connected between transcript and protein levels under DRF (Supplementary Fig. 7b).

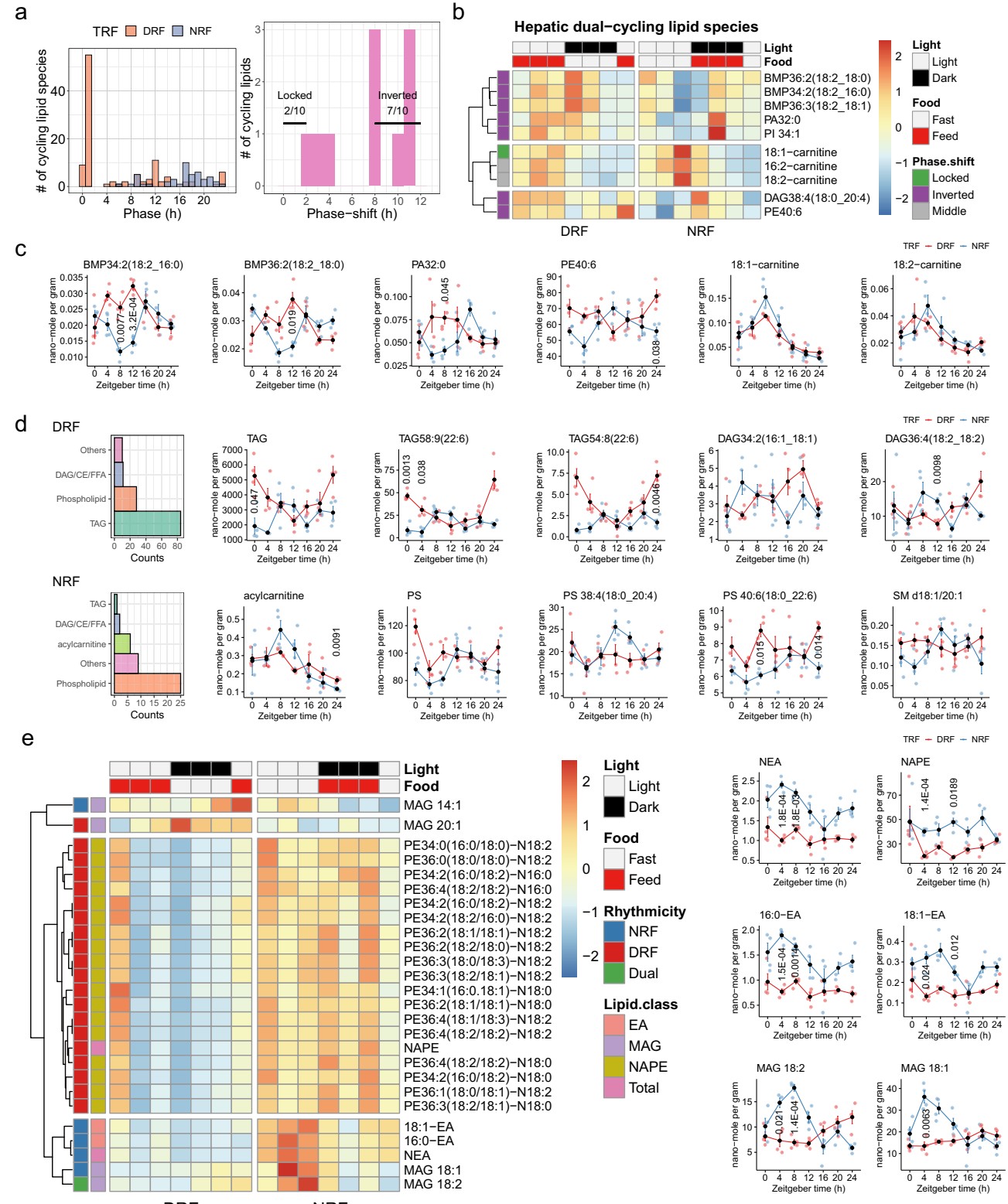

## PER2 Ser971 phosphorylation is induced by nutrient flux

We next sought to validate the nutrient-responsive activity of PER2-pSer971. Ser971 is in the carboxyl-terminal region that is involved in the physical interaction with PPARγ (Fig. 7f). AML-12 mouse hepatocytes stably expressing enhanced green fluorescence protein (EGFP)/FLAG-tagged mouse PER2, as well as the non-phosphorylatable serine-to-alanine mutant (S971A), were generated by lentiviral particles. Using a polyclonal antibody raised against the pSer971 peptide of mouse PER2 (Supplementary Fig. 7c), we showed that PER2 Ser971 is endogenously phosphorylated in hepatocytes, the signals of which were abolished in PER2[S971A] mutant (Fig. 7g). By immunoblotting analysis, we found that the levels of PER2-pSer971 increased as the concentration of extracellular glucose concentration raised from 0 to 5 mM (Fig. 7g). These data suggest that PER2-pSer971 is activated by nutrient flux in vivo.

**Fig. 6 | Diurnal lipidomics profiling corroborates the entrainment of fatty acid metabolism by DRF. a** Histograms showing phase distribution of diurnal hepatic lipids and phase-shift distribution of dual-cycling hepatic lipids, the rhythmicity of which is determined by RAIN (adjusted $P < 0.05$), MetaCycle (adjusted $P < 0.05$) and Circacompare ($P < 0.05$) ($n = 28$ mice per group, $n = 4$ mice per time point). Two-sided multiple comparisons are adjusted using the default settings. Exact $p$-values are provided in the associated Source data file. Locked, phase shift [0, 2 h]; inverted, phase shift [8, 12 h]; numbers denote phase-locked/inverted rhythmic lipids over total dual-cycling lipids. **b** Diurnal profiles of dual-cycling lipids in mouse livers ($n = 4$ mice per time point). Average expression levels were log2-transformed and scaled. BMP, bis(monoacylglycero) phosphate; DAG, diacylglycerol; PA, phosphatidic acid; PE, phosphatidylethanolamine; PI, phosphatidylinositol. **c** Diurnal

profiles of representative diurnal lipids in mouse liver. Data are presented as mean values ± standard error of the mean, $n = 4$ mice per time point for 7 time points. Two-sided unpaired Student's $t$-tests with Bonferroni correction. **d** Class of diurnal hepatic lipids and 24-h levels of representative diurnal lipids from TRF mice. TAG, triacylglycerol; PS, phosphatidylserine; SM, sphingomyelin. **e** Diurnal profiles of diurnal N-acylphosphatidyl ethanolamine (NAPE), endocannabinoid/N-acylethanolamine (NAE) and monoacylglycerol (MAG) species in TRF mouse livers, the rhythmicity of which is determined by RAIN ($P < 0.05$), MetaCycle ($P < 0.05$), and Circacompare ($P < 0.05$). Data are presented as mean values ± standard error of the mean, $n = 4$ mice per time point for 7 time points. Two-sided unpaired Student's $t$-tests with Bonferroni correction. Source data are provided as a Source data file.

## Discussion

Here, we have compiled a comprehensive multi-omics resource of daily rhythms in mouse livers under time-restricted feeding and explored their dynamics upon reversal of meal timing. 44.8% (2139/4778) of phospho-proteins, 37.8% (483/1277) of ubiquityl-proteins, 34.3% (1983/5773) of proteins, 15.7% (196/1252) of N-glycosyl proteins, and 1.9% (24/1286) of succinyl-proteins were rhythmic in the livers from TRF female mice. This is matched with 31.4% (195/622) of rhythmic lipids covering 33 classes in the liver. Integrative analysis identified clock regulation of fatty acid metabolism as a key TRF-regulated diurnal feature of the liver function. This is also corroborated by gene-level connectivity map analysis. Pathway-level connectivity mapping suggested that protein phosphorylation and N-glycosylation act as hubs in the diurnal network of liver function under NRF and DRF, respectively. Through mutagenesis study, we showed that PER2-pSer971 senses the availability of free fatty acids and glucose in vivo.

The diurnal landscape of physiology and metabolism regulated by meal timing is key to understand the mechanisms and health benefits of time-restricted feeding. While recent studies demonstrate the ready response of the liver clock and hepatic transcriptomes to DRF[11,12,14], the hepatic metabolome is hardly entrained to meal timing in female mice[14], indicating the potential role of PTMs and the lipidome in connecting meal timing to rhythmic liver function. The affinity purification-based label-free quantification LC-MS/MS platform is emerging as a powerful technique to reveal diurnal rhythms beyond the transcriptome. While a single proteomics or PTM-proteomics cannot cover the whole proteome in liver tissue, we integrated five (PTM)-proteomics that include non-modified proteins, regulatory proteins and compartmentalized proteins (i.e., mitochondria-enriched succinylation and secretory vesicle/membrane-bound N-glycosylation), and matched the proteome to the lipidome. This technical advance provides numerous details on the regulation of rhythmic liver function by meal timing.

Protein phosphorylation is a well-established PTM that couples the circadian clock to meal timing[2,6]. Glucose-sensing AMP-activated protein kinase resets the peripheral clock by phosphorylation and subsequent ubiquitin-dependent degradation of CRY1 (refs. 52,53). $O$-linked $\beta$-$N$-acetyl glucosamine ($O$-GlcNAc) glycosylation couples the circadian clock to glucose availability by inhibiting PER2 phosphorylation and inhibiting BMAL1/CLOCK ubiquitylation[54–56]. Insulin receptor signaling resets the liver clock and transcriptome during ad libitum feeding and DRF[57,58]. Along these lines, our work demonstrates that protein phosphorylation is the most clock-modulated PTM in the liver, with 44.8% of such modified proteins exhibiting diurnal rhythms in the liver in response to TRF.

In contrast, protein succinylation is the least responsive PTM in liver tissue toward changes of meal timing. We profiled rhythmic succinylated proteins and identified 7,944 succinylation sites, of which only 15 sites are entrained to DRF. This is unexpected because mitochondrial metabolism and dynamics are directly regulated by the liver clock[59]. In addition, the role of protein N-glycosylation is not appreciated in circadian biology. Our affinity purification-mass spectrometry

of the liver N-glycomes revealed that 5 (33% of the 15) were reversed in phase by DRF compared to NRF. It has been shown that 590 ubiquitylated proteins exhibit daily rhythms in synchronized human cells[60]. Our dataset shows that 483 ubiquitylated proteins entrain to meal timing in mouse livers, which illustrates the landscape of diurnal rhythms in protein ubiquitylation in vivo.

Rhythmic lipids regulate circadian rhythms of behavior and metabolism[26]. Daily release of phosphatidylcholine 18:0/18:1 from the liver modulates muscle fatty acid uptake[61]. Brain ceramide phosphoethanolamine and sphingolipids regulate the circadian clock in fruit flies[62,63]. In mammals, 30% of nuclear and mitochondrial lipids cycle robustly in the liver[33]. Considering the finding that fatty acid metabolism is a prominent proteomic feature entrained by meal timing, we profiled the hepatic lipidome as a function of meal timing. Remarkably, most lipids entrain to DRF, including BMP, TAG and phospholipids. Autophagy and the circadian clock show a bi-directional interaction[64,65]. BMP is enriched in lysosomes, the robust entrainment of which to DRF may indicate the interaction between lysosomal functions and meal timing.

Overall, we have generated a comprehensive proteomics- and lipidomics-based database (http://www.circametdb.org.cn) for diurnal rhythms in the liver under time-restricted feeding, which should shed light on clock-modulated check-points in physiology and disease.

Our study does have a few limitations. Notably, while our work provides a snapshot of the proteomic landscape of daily rhythms in the liver during time-restricted feeding, it does not cover all the known clock-regulatory PTMs. For example, methylation, SUMOylation, and $O$-GlcNAcylation were not included in the study. The depth of this multi-level proteomics could be improved with more fractionation. Furthermore, the sampling frequency could be further improved by decreasing the sampling time from 4 h to 1 h. Ideally, a 1-h 2-day sampling scheme would identify not only circadian rhythms but also ultradian rhythms. Also, the immune response signature found in the DRF proteome may indicate the diurnal activity of leukocytes, which raises the concern that the findings are based on the population average of a heterogeneous organ. This concern could be addressed by single-cell or spatial omics. Finally, this study uses solely female mice. Validation in male mice, including further mechanistic studies, is warranted to identify sex-dependent and -independent effects of meal timing on diurnal rhythms of various omics in the liver.

## Methods
### Animals
Animal experiments were approved by the Laboratory Animal Welfare and Ethics Committee of Army Medical University (AMU), China (approval no. AMUWEC20201106). All experiments conformed to the relevant regulatory standards of AMU. Special pathogen-free (SPF) mice were purchased from Hunan SJT Laboratory Animal Co. and housed in a SPF barrier facility. C57BL/6J female mice at 7 weeks of age were grouped housed and entrained to a 12 h light:12 h dark cycle (light intensity 200 lux, humidity 40–60%, ambient temperature 21–23 °C) with free access to normal chow food (Jiangsu-Xietong, SFC9112-

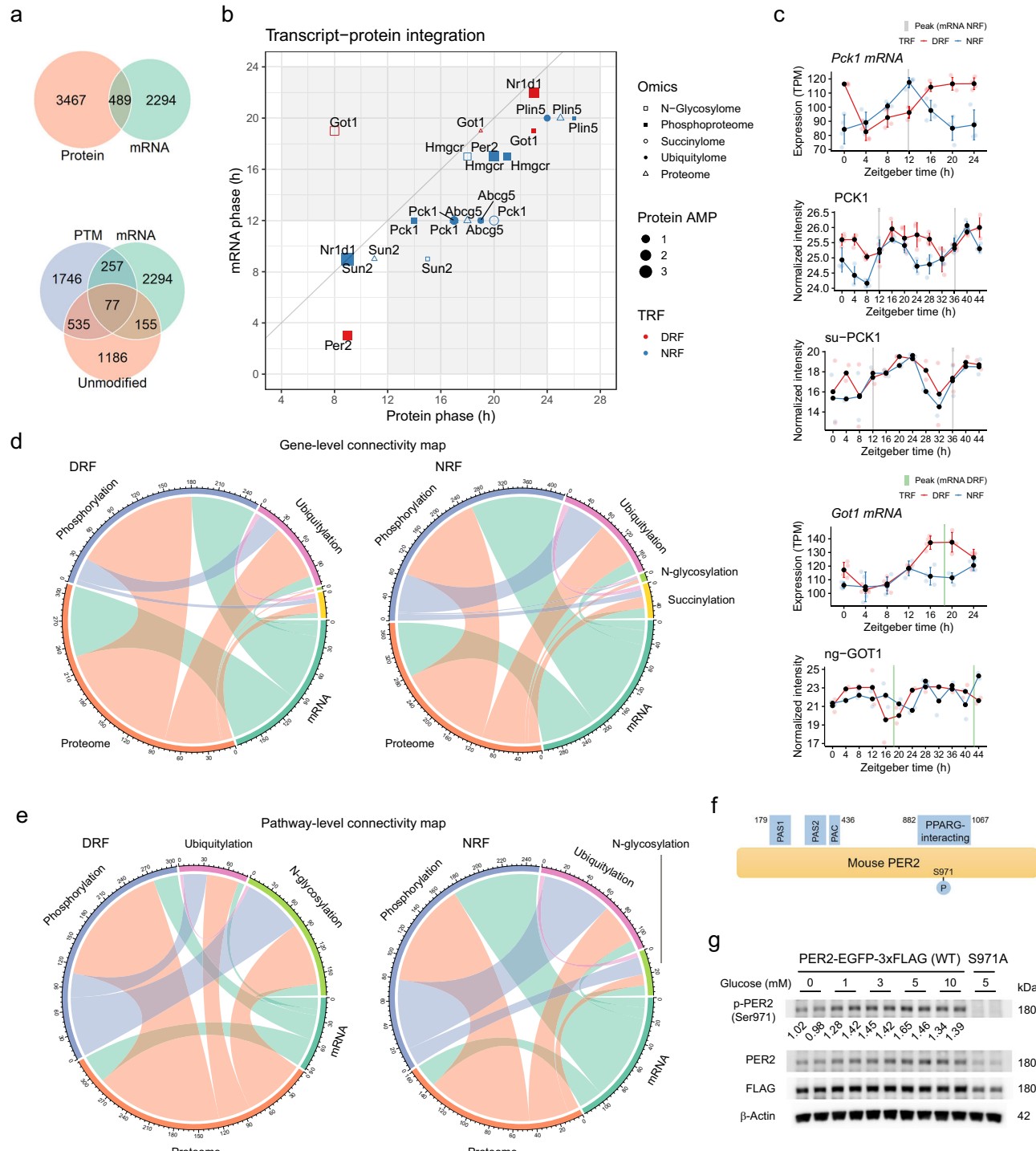

**Fig. 7 | Integrated analyses of diurnal transcriptome and multi-level proteomes. a** Venn diagram showing the interaction between rhythmic proteins consisting of unmodified, phosphorylated (phos), ubiquitylated (ubiq), N-glycosylated (ngly), and succinylated (succ) proteins and rhythmic transcripts in mouse livers under either DRF or NRF, as measured by the presence of source genes. Rhythmic transcripts are based on a published dataset (GSE150380), as measured by MetaCycle: meta2d_BH.Q < 0.05 and meta2d_rAMP >0.1 ($n = 28$ mice per group). **b** Phase plot showing the phase relationship between rhythmic proteins with or without post-translational modifications and rhythmic transcripts. **c** Diurnal profiles of *Pck1* and *Got1* gene products in mouse livers under TRF. Data are presented as mean values ± standard deviation (mRNA and protein). $N = 14$ (succinylation (su-) or N-glycosylation (ng-)) or 28 (mRNA or protein) per group. **d** Chord diagram showing the connection across multi-level proteomes and transcriptome

under DRF or NRF, as measured by the number of shared rhythmic gene products (protein or transcript). **e** Chord diagram showing the connections among different omics, as measured by the presence of rhythmic pathways (Kuiper test, $q < 0.05$). **f** A schematic diagram illustrating the structural features of mouse PER2 protein. PAS, Per-ARNT-Sim domain; PAC, PAS-associated C-terminal domain; PPARG, peroxisome proliferator-activated receptor gamma. **g** Representative immunoblots of PER2-pSer971 and PER2 in AML-12 mouse hepatocytes stably expressing EGFP/FLAG-tagged PER2 or S971A mutant. Cells were treated in different concentrations of glucose for 8 h ($n = 8$ biological replicates from 4 independent experiments). Densitometric results were subtracted from the average value of those from S971A, normalized to the average of the 0 mM glucose group and labeled below the bands. Source data are provided as a Source data file.

1010083, 22.8% kcal protein, 13.8% fat and 63.4% kcal carbohydrate, 3656 kcal/kg) and water.

## Time-restricted feeding

C57BL/6J female mice at 7 weeks of age were grouped housed and entrained to a 12 h light:12 h dark cycle (light intensity 200 lux) with normal chow food and water ad libitum for at least 7 days before assigned to daytime-restricted feeding (DRF) and nighttime-restricted feeding (NRF) groups for 7 days. To synchronize the estrus cycle (4–5 days) of our animals, we co-housed these mice in a big rodent cage for 5–7 days before relocating to regular cages ($n = 4$ per cage) in our SPF facility. When assigning the mice to different groups, we randomized and controlled the body weight statistics to make sure the body weight is comparable between NRF and DRF groups. The DRF group had access to food for 12 h from ZT0 to ZT12 where ZT0 denotes light on at 9:00 am in China Standard Time (UTC + 8). The NRF group had access to food for 12 h from ZT12 to ZT24 (ZT0). At the end of time-restricted feeding, mice were euthanized by cervical dislocation and subjected to tissue collection every 4 h over 24 h (lipidomics) or 48 h (proteomics and PTM proteomics). Liver tissue was dissected ($n = 4$ per group per time-point totaling 56 or 96 samples). Tissues were snap frozen in liquid nitrogen and stored in a −80 °C fridge.

## Protein extraction

Mouse liver samples were pulverized in liquid nitrogen and sonicated in the urea lysis buffer containing 100 mM $NH_4HCO_3$ (pH 8), 8 M Urea and 0.2% Sodium dodecyl sulfate (SDS) for 5 min on ice. The lysate was centrifuged at $12,000 \times g$ for 15 min at 4 °C, the supernatants were reduced by 10 mM DTT for 1 h at 56 °C, and subsequently alkylated with sufficient iodoacetamide (Sigma, #I6125-25G) for 1 h at room temperature in the dark. Next, samples were completely mixed with 4X volume of precooled acetone (Beijing Chemical Works, #11241203810051) by vortexing and incubated at −20 °C for at least 2 h. After centrifugation, the pellet was washed twice with cold acetone, and dissolved by dissolution buffer, which containing 0.1 M triethylammonium bicarbonate (TEAB, pH 8.5; Sigma, #T7408-500ML) and 6 M urea. Protein samples were subjected to 12% SDS-PAGE, all of which showed no visible sign of degradation.

## Proteomics, DIA

120 μg liver proteins were aliquoted for peptide digestion by trypsin (Promega, #V5280), stopped with equal volume of 1% formic acid (HCOOH, Fisher Chemical, #A117-50), desalted in a C18 column and lyophilized. Lyophilized samples were dissolved in mobile phase A (2% acetonitrile or ACN, Fisher Chemical, #A955-4; adjusted to pH 10.0 by ammonium hydroxide) and fractionated in a C18 column (Waters BEH C18 4.6 × 250 mm, 5 μm) on a Rigol L3000 HPLC system at 50 °C (column oven) in an elution gradient mixed by mobile phase A and B (98% ACN, pH 10.0) at a flow rate of 1 mL/min as follows. 3–5% B, 10 min; 5–20% B, 20 min; 20–40% B, 18 min; 40–50% B, 2 min; 50–70% B, 3 min; 70–100% B, 1 min. Elutes were combined into 10 fractions, vacuum-dried and reconstituted in 0.1% formic acid in $H_2O$.

For data-dependent acquisition (DDA) spectrum library construction, shotgun proteomics analyses were performed using an EASY-nLC 1200 UHPLC system (Thermo Fisher, #LC140) coupled to an Q Exactive HF-X mass spectrometer (Thermo Fisher) via a nano-electrospray ion source (Nanospray Flex, spray voltage of 2.5 kV) at a capillary temperature of 320 °C. 1 μg sample containing iRT standard peptides (Biognosys, #Ki-3002-2) was analyzed by UHPLC at a flow rate of 600 nL/min in a linear gradient as follows. 5–8% B, 1 min; 8-30% B, 75 min; 30–50% B, 5 min; 50–95% B, 1 min; 95% B, 10 min; 95–5% B, 0.5 min; 5% B, 1 min; 5–95% B, 0.5 min; 95% B, 5 min; 95–5% B, 1 min. The Q Exactive HF-X was operated in Top40 mode with a full scan of 350–1500 m/z at a resolution of 120,000 ($m/z$ 200). The automatic

gain control (AGC) target value was set to $3 \times 10^6$ at a maximum ion injection time of 80 ms. Precursor ions were fragmented by higher energy collisional dissociation (HCD) at a normalized collision energy (NCE) of 27% and analyzed in MS/MS, with the resolution set to 15,000 ($m/z$ 200), an AGC of $5 \times 10^4$, a maximum ion injection time of 45 ms, an intensity threshold of $1.1 \times 10^4$ and a dynamic exclusion parameter of 20 s. For DIA acquisition, the m/z range was set to 350–1500 $m/z$ at a resolution of 60,000 ($m/z$ 200). The AGC was set to $5 \times 10^5$ at a maximum ion injection time of 20 ms. Peptides were fragmented by HCD (NCE 27%) and fragment ions were analyzed in 50 scan windows in MS2, with the resolution set to 30,000 (200 $m/z$) and the AGC target value set to $1 \times 10^6$.

DDA and DIA data were analyzed using Proteome Discoverer (Thermo, v2.4), Spectronaut (Biognosys, v9.0), and R against the UniProt mouse reference proteome (Proteome ID UP000000589, release 2019.01.18). DDA MS raw files were analyzed by Proteome Discoverer, and peak lists were searched against protein database. Cysteine carbamidomethylation (C/ + 57.021 Da) was set as a fixed modification; N-terminal acetylation (N-terminal/+42.011 Da) as N-Terminal modification; methionine oxidation (M/ + 15.995 Da) as dynamic modification. In peptide identification, maximum missed cleavage sites were set to 2, precursor mass tolerance set to 10 ppm and fragment mass tolerance set to 10 ppm. The false discovery rate (FDR) was set to 1% for both proteins and peptides, and was determined by searching a protein reverse database. MS1 label-free quantification (LFQ) was analyzed by maxLFQ algorithm and MS2 LFQ of DIA raw data was analyzed by Spectronaut with the following parameters, i.e., data extraction was set to "dynamic" (correction factor 1), identification set to "normal distribution p-value estimator" ($q$-value < 0.01), profiling strategy set to "iRT profiling" ($q$-value < 0.01), protein inference set to "from search engine", protein quantity set to "average precursor quantity" and smallest quantitative unit set to "precursor ion" (summed fragment ions).

## Phosphoproteomics, DDA

5 mg proteins were aliquoted for trypsin digestion, desalted in the C18 desalting column and lyophilized. Phospho-peptide fractions were enriched in an IMAC-Fe column (Fisher Scientific, #A32992) and incubated at room temperature for 30 min before centrifugation and lyophilization. Shotgun proteomics analyses were performed using an EASY-nLC 1200 UHPLC system (Thermo Fisher) coupled with an Q Exactive HF-X mass spectrometer (Thermo Fisher) operating in the DDA mode via a nano-electrospray ion source (Nanospray Flex, spray voltage of 2.3 kV) at a capillary temperature of 320 °C. 1 μg sample was analyzed by UHPLC at a flow rate of 600 nL/min in a 120 min linear gradient from 5 to 100% of eluent B (0.1% HCOOH in 80% ACN) in eluent A (0.1% HCOOH in $H_2O$) at a flow rate of 600 nL/min and detailed as follows, 5–10% B, 2 min; 10–30% B, 110 min; 30–50% B, 5 min; 50–95% B, 1 min; 95% B, 5 min. The Q Exactive HF-X was operated in Top30 mode with a full scan of 350–1500 $m/z$ at a resolution of 120,000 ($m/z$ 200). The AGC target value was set to $3 \times 10^6$ at a maximum ion injection time of 80 ms. Precursor ions were fragmented by HCD at a NCE of 27% and analyzed in MS/MS, with the resolution set to 15,000 ($m/z$ 200), an AGC of $5 \times 10^4$, a maximum ion injection time of 100 ms, an intensity threshold of $5 \times 10^3$ and a dynamic exclusion parameter of 30 s.

The resulting spectra were searched against the UniProt mouse reference proteome (Proteome ID UP000000589, release 2019.01.18) in Proteome Discoverer (v.2.2). Searched parameters: maximum missed cleavage sites of 2, mass tolerance of 10 ppm for precursor ion scans and a mass tolerance of 0.02 Da for the product ion scans. Cysteine carbamidomethylation (C/+57.021 Da) was set as a fixed modification; N-terminal acetylation (N-terminal/+42.011 Da) as N-Terminal modification; phosphorylation of serine, threonine and tyrosine residue (S, T, Y/+79.966 Da) and methionine oxidation

(M/+15.995 Da) as dynamic modifications; FDR < 1% at levels of PSM and protein (matched with at least one unique peptide).

## Ubiquitylomics, DDA

10 mg proteins were aliquoted for trypsin digestion, and incubated with anti-Ubiquitin Remnant Motif (K-ε-GG) beads (CST #5562) in MOPS IAP buffer (50 mM MOPS, 10 mM $KH_2PO_4$, and 50 mM NaCl) for 2 h at 4 °C. Shotgun proteomics analyses were performed using an EASY-nLC 1200 UHPLC system (Thermo Fisher) coupled with an Q Exactive HF-X mass spectrometer (Thermo Fisher) operating in the DDA mode via a nano-electrospray ion source (Nanospray Flex, spray voltage of 2.3 kV) at a capillary temperature of 320 °C. 1 µg sample was analyzed by UHPLC at a flow rate of 600 nL/min in a 120 min linear gradient from 5 to 100% of eluent B (0.1% HCOOH in 80% ACN) in eluent A (0.1% HCOOH in $H_2O$) at a flow rate of 600 nL/min and detailed as follows, 5–10% B, 2 min; 10–30% B, 110 min; 30–50% B, 5 min; 50–95% B, 1 min; 95% B, 5 min. The Q Exactive HF-X was operated in Top30 mode with a full scan of 350–1500 $m/z$ at a resolution of 120,000 (200 m/z) with an AGC target value of $3 \times 10^6$ and a maximum ion injection time of 80 ms. Precursor ions were fragmented by HCD at a NCE of 27% and analyzed in MS/MS, with the resolution set to 15,000 ($m/z$ 200), an AGC of $5 \times 10^4$, a maximum ion injection time of 100 ms, an intensity threshold of $5 \times 10^3$ and a dynamic exclusion parameter of 30 s.

The resulting spectra were searched against the UniProt mouse reference proteome (Proteome ID UP000000589, release 2020.01.08) in Proteome Discoverer (v.2.4). Searched parameters: maximum missed cleavage sites of 2, mass tolerance of 10 ppm for precursor ion scans and a mass tolerance of 0.02 Da for the product ion scans. Cysteine carbamidomethylation (C/ + 57.021 Da) was set as a fixed modification; N-terminal acetylation (N-terminal/+42.011 Da) as N-Terminal modification; GG of lysine (K/ + 114.043 Da) and methionine oxidation (M/ + 15.995 Da) as dynamic modification; FDR < 1% at levels of PSM and protein (matched with at least one unique peptide).

## Succinylomics, DDA

10 mg proteins were aliquoted for trypsin digestion, and incubated with anti-succinyl-lysine motif beads (CST #13764) in MOPS IAP buffer for 2 h at 4 °C. Shotgun proteomics analyses were performed using an EASY-nLC 1200 UHPLC system (Thermo Fisher) coupled with an Q Exactive HF-X mass spectrometer (Thermo Fisher) operating in the DDA mode via a nano-electrospray ion source (Nanospray Flex, spray voltage of 2.3 kV) at a capillary temperature of 320 °C. 1 µg sample was analyzed by UHPLC at a flow rate of 600 nL/min in a 120 min linear gradient from 5 to 100% of eluent B (0.1% HCOOH in 80% ACN) in eluent A (0.1% HCOOH in $H_2O$) at a flow rate of 600 nL/min and detailed as follows, 5–10% B, 2 min; 10–30% B, 110 min; 30–50% B, 5 min; 50–95% B, 1 min; 95% B, 5 min. The Q Exactive HF-X was operated in Top30 mode with a full scan of 350–1500 $m/z$ were acquired at a resolution of 120,000 (200 $m/z$) with an AGC target value of $3 \times 10^6$ and a maximum ion injection time of 80 ms. Precursor ions were fragmented by HCD at a NCE of 27% and analyzed in MS/MS, with the resolution set to 15,000 (200 $m/z$) with an AGC target value of $5 \times 10^4$, a maximum ion injection time of 100 ms, an intensity threshold of $5 \times 10^3$ and the dynamic exclusion parameter of 30 s.

The resulting spectra were searched against the UniProt mouse reference proteome (Proteome ID UP000000589, release 2020.01.08) in Proteome Discoverer (v.2.4). Searched parameters: maximum missed cleavage sites of 2, mass tolerance of 10 ppm for precursor ion scans and a mass tolerance of 0.02 Da for the product ion scans. Cysteine carbamidomethylation (C/ + 57.021 Da) was set as a fixed modification; N-terminal acetylation (N-terminal/+42.011 Da) as N-Terminal modification; succinylation of lysine (K/ + 100.016 Da) and methionine oxidation (M/ + 15.995 Da) as dynamic modification; FDR < 1% at levels of PSM and protein (matched with at least one unique peptide).

## N-Glycomics, DDA

500 µg proteins were aliquoted for trypsin digestion, desalted, and vacuum-dried. Peptides were dissolved in 80% ACN (0.1% trifluoracetic acid) and enriched for N-glycosylated peptides in the HILIC column (Bonna-Agela, #VH950010). The mix was first washed in 0.1% trifluoracetic acid, then in 50 mM $NH_4HCO_3$-$H_2^{18}O$ and digested in PNGase-F (Sigma, #P7367) overnight at room temperature. Shotgun proteomics analyses were performed using an EASY-nLC 1200 UHPLC system (Thermo Fisher) coupled with an Q Exactive HF-X mass spectrometer (Thermo Fisher) operating in the DDA mode via a nano-electrospray ion source (Nanospray Flex, spray voltage of 2.3 kV) at a capillary temperature of 320 °C. 1 µg sample was analyzed by UHPLC at a flow rate of 600 nL/min in a 120 min linear gradient from 5 to 100% of eluent B (0.1% HCOOH in 80% ACN) in eluent A (0.1% HCOOH in $H_2O$) at a flow rate of 600 nL/min and detailed as follows, 5–10% B, 2 min; 10–40% B, 105 min; 40–50% B, 5 min; 50–90% B, 3 min; 90–100% B, 5 min. The Q Exactive HF-X was operated in Top40 mode with a full scan of 350–1500 $m/z$ with a resolution of 60,000 (200 $m/z$) with an AGC target value of $3 \times 10^6$ and a maximum ion injection time of 20 ms. Precursor ions were fragmented by HCD at a NCE of 27% and analyzed in MS/MS, with the resolution set to 15,000 (200 $m/z$) with an AGC target value of $5 \times 10^4$, a maximum ion injection time of 80 ms, an intensity threshold of $1.3 \times 10^4$, and the dynamic exclusion parameter of 30 s.

The resulting spectra were searched against the UniProt mouse reference proteome (Proteome ID UP000000589, release 2019.01.18) in Proteome Discoverer (v.2.2). Searched parameters: maximum missed cleavage sites of 2, mass tolerance of 10 ppm for precursor ion scans and a mass tolerance of 0.02 Da for the product ion scans. Cysteine carbamidomethylation (C/ + 57.021 Da) was set as a fixed modification; N-terminal acetylation (N-terminal/+42.011 Da) as N-Terminal modification; glycosylation of asparagine (deamidation $^{18}O$ (N)/ + 2.988 Da) and methionine oxidation (M/ + 15.995 Da) as dynamic modification; FDR < 1% at levels of PSM and protein.

## Targeted lipidomics

Lipids were extracted from approximately 20 mg tissues using a modified version of the Bligh and Dyer's method[34]. Liver tissues were homogenized in 750 µL of chloroform:methanol 1:2 (v/v) with 10% deionized water on a bead ruptor (OMNI, USA). The homogenate was then incubated at 1500 rpm for 1 h at 4 °C. At the end of the incubation, 350 µL of deionized water and 250 µL of chloroform were added to induce phase separation. The samples were then centrifuged and the lower organic phase containing lipids was extracted into a clean tube. Lipid extraction was repeated once by adding 500 µL of chloroform to the remaining tissues in aqueous phase, and the lipid extracts were pooled into a single tube and dried in the SpeedVac under OH mode. Polar lipids were analyzed using an Exion-UPLC system coupled with a triple quadrupole/ion trap mass spectrometer (6500 Plus Qtrap; SCIEX)[35,66]. Separation of individual lipid classes of polar lipids by normal phase (NP)-HPLC was carried out using a Phenomenex Luna 3 µm-silica column (internal diameter 150 × 2.0 mm) with the following conditions: mobile phase A (chloroform:methanol:ammonium hydroxide, 89.5:10:0.5) and mobile phase B (chloroform:methanol:ammonium hydroxide:water, 55:39:0.5:5.5). MRM transitions were set up for comparative analysis of various polar lipids. Glycerol lipids including diacylglycerols and triacylglycerols were quantified using a modified version of reverse phase HPLC/MRM. Separation of neutral lipids were achieved on a Phenomenex Kinetex-C18 2.6 µm column (4.6 × 100 mm) using an isocratic mobile phase containing

chloroform:methanol:0.1 M ammonium acetate 100:100:4 (v/v/v) at a flow rate of 170 μL for 17 min.

## Targeted profiling of NAE and NAPE

N-acylethanolamines (NAEs) and N-acyl phosphatidylethanolamines (NAPEs) were extracted from liver tissues using a modified method of Bligh and Dyer's extraction[67]. The organic phase was extracted and dried in SpeedVac under organic mode. Samples were resuspended in acetonitrile: isopropanol (1:2 v/v) containing appropriate concentrations of internal standards including d7-PE33:1 from Avanti Polar Lipids, d8-20:4-EA, d4-16:0-EA, d4-22:6-EA and d5-MAG-20:4 from Cayman Chemicals. Samples were analyzed on a ThermoFisher U3000 DGLC coupled to Sciex 6500 Plus QTRAP under the electrospray ionization mode. Individual lipids were separated on an Agilent Zorbax Eclipse Plus column (100 × 2.1 mm, 1.8 μm) using Mobile Phase A (10 mM ammonium formate: acetonitrile: isopropanol 50:30:20, pH 8) and Mobile Phase B (10 mM ammonium formate: acetonitrile: isopropanol 1:9:90, pH 8). NAEs and NAPEs were quantitated by referencing to spiked internal standards.

## DIABLO integrative analysis

Integrative network analysis of the hepatic phosphoproteome, proteome, and ubiquitylome was performed using the DIABLO (Data Integration Analysis for Biomarker discovery using Latent variable approaches for Omics studies) model in the R package mixOmics[47]. First, omics features were selected using sparse Partial Least Squares Discriminant Analysis (sPLS-DA) based on three subsets of input data, i.e., morning (ZT0-4), evening (ZT8-12) or night (ZT16-20). Abundance profiles of 2139 rhythmic phosphoproteins, 1983 rhythmic proteins, and 483 rhythmic ubiquitylated proteins found in livers from either DRF or NRF mice served as input data. The evening subset was chosen for modeling study for its low BER. Specifically, the following modeling parameters associated with the evening subset were used after tuning: ncomp = 3 (number of model components); the number of rhythmic proteins to consider: 20; the number of rhythmic phospho-proteins to consider: 6; the number of rhythmic ubiquitylated proteins to consider: 5, 5. Data were then modeled in the DIABLO framework. Plots of experimental groups (function: plotIndiv) provide a visual representation of samples in the DIABLO framework. Correlation Circle plot (function: circosPlot), Clustered Image Maps for DIABLO (function: cimDiable), and plots of variable loadings (importance of the feature/molecule) were produced in the mixOmics R package.

## Cell culture and chemicals

Alpha mouse liver 12 (AML-12) cell line was established from mouse hepatocytes (source: a 3-month-old CD1 strain MT42 mouse line) that ectopically express human transforming growth factor alpha (ATCC #CRL-2254). AML12 cells were authenticated by their general morphology, short tandem repeat microsatellite marker analysis, capacity to form lipid droplets upon oleic acid treatment and RT-qPCR for mouse genes. AML-12 cells were cultured at 37 °C and 5% $CO_2$ in DMEM/F12 (Procell #PM150312) supplemented with 10% fetal bovine serum (Procell #164210-50), 10 μg/mL insulin, 5.5 μg/mL transferrin, 5 ng/mL selenium, 40 ng/mL dexamethasone and 1% penicillin/streptomycin (100 U/mL and 0.1 mg/mL, respectively; Procell #PB180120), which is formulated as the AML-12 culture medium (Procell #CM-0602). Cells were seeded to 6-well plates (NEST #703001) or 10-cm dish (NEST #704002). When reaching 80-90% confluence, cells were treated in DMEM (Gibco #11966025) containing various concentrations of glucose (BioReagent grade, Sigma #G7021) for 8 h.

## Generation of lentiviral particles and stable cell lines

Mouse *Per2* cDNA (NM_011066.3, CDS 175..3948 nucleotides) or its S971A variant was generated by a Gibson Assembly Cloning Kit (NEB

#E5510) and cloned into the lentiviral vector pLenti-hEF1a-PuroR-CMV-MCS-EGFP-3xFlag (BamHI/XmaI). Lentiviral particles, i.e., Lenti-hEF1a-PuroR-CMV-*Per2*-EGFP-3xFlag (3.44E + 08 transduction unit per mL or TU/mL) and Lenti-hEF1a-PuroR-CMV-*Per2*<sup>S971A</sup>-EGFP-3xFlag (3.76E + 08 TU/mL) were prepared by the vendor (Shanghai Tailtool Bioscience Co. Ltd). AML-12 mouse hepatocytes in early passages were seeded at 10,000 cells per well into a 6-well cell culture microplate and transduced with 5 μg/mL polybrene and lentiviral particles at a multiplicity of infection (MOI) of 50 over the night. These cells express PER2 or PER2<sup>S971A</sup> as a fusion protein that is tagged in the carboxyl-terminal with EGFP and 3xFLAG peptide. Stable cells were cultured at 5 μg/mL puromycin (Beyotime #ST551) for 7–10 days and selected for EGFP biofluorescence under a Nikon-U fluorescence microscope.

## Immunoblot analysis

Cells were lysed in RIPA buffer (ThermoFisher #89901) supplemented with proteinase inhibitor cocktail (Roche #5892953001) and protein phosphatase inhibitor cocktail (PhosSTOP, Roche #4906845001). 15 μg of protein lysates were electrophoresed on SDS-PAGE gels at 110 V and transferred to activated PVDF membranes (Bio-Rad #1620177). Block the membranes in 5% skim milk or 5% BSA solution (PER2-pSer971) for 1.5 h and incubated with primary antibodies at a dilution of 1:1000 (except β-Actin 1:50,000) and 4 °C overnight. PER2 rabbit polyclonal antibody (pAb) (Abclonal #A13168, RRID:AB_2760019, 1:1000); PER2-pSer971 rabbit pAb (Abclonal #WG-00762P, 1:1000); β-Actin rabbit mAb (Abclonal #AC026, RRID:AB_2768234, 1:50,000); DYKDDDDK (FLAG) tag rabbit pAb (Proteintech, 20543-1-AP, RRID: AB_11232216, 1:1000). Immunoblots were visualized by peroxidase conjugated secondary antibodies, i.e., HRP-linked anti-mouse IgG (Cell Signaling Technology, Cat# 7076, RRID: AB_330924, 1:5000) and HRP-linked anti-rabbit IgG (Cell Signaling Technology, Cat# 7074, RRID: AB_2099233, 1:5000), and enhanced chemiluminescent substrate (Bio-Rad #1705062) in the Azure C500 imaging system (see Fig. 7g and associated Source Data file). PVDF membranes were stripped in antibody removal solution (Beyotime #P0025B) for 20 min and washed three times in TBST before re-starting the blocking and immunoblotting procedure. Experiments were reproduced with similar results ($n = 8$ biological replicates from 4 independent experiments). PER2-pSer971 rabbity polyclonal antibody was generated by Abclonal Inc. Phosphopeptide GRA(S-p)PPLFQSR-C (mPER2 968-978 AA) was conjugated to KLH and immunized to SPF rabbits (*Oryctolagus cuniculus*). The antibody was purified from sera by affinity purification and screened for positive signals against the phosphopeptides and negative signals against the control peptide GRASPPLFQSR-C.

## Diurnal rhythmicity analysis

**Label-free proteomics.** Raw quantification from label-free proteomics were filtered for missing values in <30% of samples, log2-transformation and median group normalization. Within group Pearson correlation coefficients were examined and no sample was excluded. Protein matrix was then analyzed by MetaCycle::meta2d function, RAIN::rain and Circacompare::circacompare (alpha_threshold = 1) for a period length of 24 h and a time interval of 4 h.

**Lipidomics.** Quantification data in absolute units per group (treatment - time point) were checked for Pearson's correlation. All samples exhibited a r > 0.8 per group, and no sample was excluded for diurnal analysis. Lipidomics data were then analyzed by MetaCycle::meta2d function, RAIN::rain and Circacompare::circacompare (alpha_threshold = 1) for a period length of 24 h and a time interval of 4 h.

## Phase set enrichment analysis

Phase set enrichment analysis was performed on oscillating genes in different tissues to identify phase-clustered pathways via PSEA1.1_VectorGraphics.jar[68]. Source files (v7.1) are obtained from

the Molecular Signatures Database (Broad Institute, MA, USA). Mouse gene nomenclature was converted to human gene nomenclature via "Human and Mouse Homology Classes with Sequence information" (curated by http://www.informatics.jax.org). Parameters: domain 0 to 24, minimal number 10, and maximal number 10000. $Q$-value < 0.05 is considered as statistical significance. Results were sorted by Vector-average magnitude and Vector-average value, respectively before visualization.

## Statistics

Sample size was determined by Guidelines for Genome-Scale Analysis of Biological Rhythms[69]. Rhythmic proteins, sites or lipids were considered as reaching statistical significance in three rhythmicity methods, i.e., meta2d_pvalue < 0.05 (MetaCycle)[46], adjusted $P$-value < 0.05 (RAIN)[44] and "Presence of rhythmicity $p$-value < 0.05" (Circacompare)[45]. Two-sided unpaired Student's $t$-test with or without Bonferroni correction and two-sided unpaired Wilcoxon test were performed in R (version 4.1.3)/RStudio (2023.03.0 Build 386) as indicated, and $P$-value < 0.05 is considered as statistical significance. Heatmap of diurnal expression profile was visualized by R package: pheatmap (version 1.0.12). Data were plotted via R packages: ggplot2 (version 3.3.6) and ggpubr (version 0.4.0). Venn and chord diagrams were plotted via R package: VennDiagram (version 1.7.3) and Circlize (version 0.4.15).

## Reporting summary

Further information on research design is available in the Nature Portfolio Reporting Summary linked to this article.

## Data availability

The mass spectrometry proteomics data generated in this study have been deposited to the ProteomeXchange Consortium via the PRIDE[70] partner repository under accession code PXD037604, PXD037761, PXD038276, PXD038336, PXD038481. Visualization of this dataset can be found at http://www.circametdb.org.cn. Raw quantification data are deposited to Science Data Bank under accession code https://doi.org/10.57760/sciencedb.08708 and to Figshare[71] under accession code https://doi.org/10.6084/m9.figshare.23258678. Any other relevant data are available from the corresponding author (Min-Dian Li) upon request. Source data are provided with this paper.

## Code availability

Custom code is deposited to Science Data Bank (https://doi.org/10.57760/sciencedb.08708).

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

## Acknowledgements

This work was supported by Chongqing Science Fund for Distinguished Young Scholars (CSTB2022NSCQ-JQX0009 to M.-D.L.), National Natural Science Foundation of China Grants (92057109 to M.-D.L., 32271208 to M.-D.L., 81900776 to M.-D.L.) and the Opening Fund of NHC Key Laboratory of Chronobiology (Sichuan University, NHCC-2022-02 to M.-D.L.).

## Author contributions

M.-D.L. conceptualized and designed the study. M.-D.L. acquired the funding. M.-D.L., Z.Z., G.S., and F.D. coordinated and supervised the investigation. M.-D.L., R.H., J.C., and H.X. established the methods and conducted the investigation and bioinformatics. S.L. and G.S. performed the lipidomics. M.Z., X.J., and J.L. validated cell-based experiments. M.-D.L., Z.Z., G.S., R.H., J.C., M.Z., and H.X. drafted the manuscript. All authors edited and approved the manuscript.

## Competing interests

The authors declare no competing interests.
