## [Peer Review File · Nature Communications]

REVIEWER COMMENTS

Reviewer #1 (Remarks to the Author):

This paper describes post-translational modifications in the liver shaped by meal timing. The paper is rather a resource than a scientific investigation on a specific question. The only experimental approach on Per2 is rather preliminary.

Reviewer #2 (Remarks to the Author):

In the manuscript by Huand et al., the authors have looked at the proteomics profiles of the hepatic proteins to understand their role and the role of PTMs in regulating circadian rhythm. They also correlate the proteomics data to metabolomic data to discover the effect of PTMs on hepatic energy metabolism during time restricted feeding. The proteomics methodology applied appears standard. However, it'd be helpful to include more details, such as instrument method used for phosphoproteomics experiment. Was it any different from the DDA and DIA used for non-enriched samples? Similarly, how was the data analysis different in Proteome Discoverer for phosphoproteomics, ubiquitylated or succinylated datasets? What parameters and search engines were used to detect the various post translational modifications. More details will help the reader understand the data better and it also increases confidence if they try to reproduce the methodology at their end.

Reviewer #3 (Remarks to the Author):

1) Noteworthy results

Huang et al. provide the most comprehensive characterization of Post-translational modifications in the liver during the circadian rhythm. The authors provide 4 PTM (phosphorylation, ubiquitylation, succinylation, and N-glycosylation) in two time-restricted feeding frameworks.

From such systematic characterization, key results are:

- (i) the variation of each of the PTM in the circadian rhythm,**
- (ii) the effects of feeding frameworks.**
- (iii) there are many relevant features identified, and a validation experiment was conducted for one of them, "phosphorylation of the circadian repressor PER2 at residue Ser971"**

2) Overall

In general, the manuscript depicts a relevant data collection for understanding circadian rhythm. The authors have made a clear effort to provide an organized analysis and, to a certain extent, integrate all omics.

However, while the results show key messages, I read it mainly as a resource paper. However, this is my point of view.

3) Significance to the field

The data-set will definitely be of value as a resource. The results provided are meaningful overall but will have a limited impact on the field.

4) Support of conclusions and claims

Authors support their claims.

5) Data analysis

In general, the analysis "seems to be" sound; however – from my point of view – the manuscript will benefit from certain clarifications.

4.1 Sample exclusion in the analysis:

a) For some data-types, e.g., "Label-free proteomics", samples were removed if their correlation with other samples were below a certain threshold. "N04_1 was removed from phosphoproteomics analysis. N24_4, D16_2, D24_1, and D28_3 were removed from ubiquitylproteomics analysis." How can you be certain the variability observed (related to low correlation) is not biological? A better – beyond a correlation-based – criteria would be required.

b) When conducting the MixOmics analysis: do you consider only samples with all profiles available? How this affects the analysis?

4.2 Feature selection:

There are several methods mentioned for MetaCycle::meta2d function, RAIN::rain, and Circacompare::circacompare.

a. Please clarify the selection criteria: from the reading, I understand a feature is selected only if it is identified as significant by the three methods.

b. Data is collected over 48 hours, but then in the Results section is mentioned the analysis for 48 hours, and in the Methods section, it is mentioned a 24-hour based analysis: can you clarify? Overall I would rephrase the question: please clarify when do you use a 24-hour analysis and when 48 hours?

4.3 Integration with other publicly available omic features.

While PTM is systematically characterized, and there is mention of transcriptomic analysis (some of them by the same team in the interesting paper of iScience 2021), I am missing a data-driven comparison between both omic beyond the mentions in the manuscript. Something along Protein vs. gene transcripts, or at pathway level, would help to visualize at what level both omics are coordinated or not.

6) Small modifications

Figure 1 Panel a: Please modify the figure. As it is shown, it seems to imply (from my point of view) that NRF is associated with 48h PTM and DRF to the 24h-Lipidomics. I understand it is not, but it will help to modify the panel to clarify that.

Reviewer #4 (Remarks to the Author):

In this manuscript, Huang, Xin, Zhou, Chen et al report a data-intensive study combining proteomics both with and without post-translational modifications, lipidomics and some metabolomics regarding the effect of food timing on rhythmicity of proteins and small molecules. A large part of the manuscript is a relatively descriptive compendium of measurements where the authors find that there is an over-representation of lipid metabolism in the proteomics data, and on this basis perform a lipidomics study. A slightly more mechanistic examination of PER2 Ser971 phosphorylation is presented near the end of the manuscript.

While the manuscript is well written, there is a lot of data to assimilate, and other than gross conclusions (such as the over-representation of lipid data in the proteomics experiments) it is difficult to connect the various data types. Nevertheless, this is a tremendous effort, and I'm certain the details and connections will be valuable to the scientific community.

Specific comments for improvement:

1. Given the importance and sexual dimorphism in circadian studies, I applaud the authors study of female mice in their experiments. I was however not able to find complete support for the statement that "female mice exhibit a greater consolidation of circadian rhythms of peripheral clock oscillations during DRF compared to males" on lines 89/90 of p3, with a reference to one of their previous studies (Xin et al, iScience, 2021). The iScience article

states that "Together, the liver clock and part of the adipose clock maintained similar responses to inverted feeding between male and female mice; however, peripheral clocks in kidney and heart seemed to be less robust under DRF in males." It is important to be more precise in describing the premise of this study, given that the liver clock is not strongly impacted by DRF in the authors' previous work and is the only organ studied here.

2. The mice used in the study appear to be relatively young at 7 weeks. Is there a particular reason to use animals of this age? What is the status of the estrus cycle in animals this age, and could there be any impact on the results?

3. The method used to assess rhythmicity is not clearly described, which is an important component of the paper. The authors report using three methods (RAIN, CircaCompare, Metacycle) and that "Adjusted p-values were optimized by inspecting the visual fitness in the exploratory Nitecap platform." The methods have more specific details on the cutoffs used (which is appreciated, and the fact these are available in the methods should be referenced in the main text with the previous sentence). At the same time, it is a bit disconcerting to see cut-offs based on somewhat arbitrary visual inspection so this needs to be better rationalized.

4. I am a little confused by the definition of dual-cycling. For example, the legend for Figure 6a indicates the phase shift for dual-cycling lipids, which I assume to be a histogram. Counting the number of samples, there appears to be a total of 10 lipids, which is consistent with the heatmap of 6b. However, the percentages reported do not make sense-- P11, lines 370/371: "Strikingly, 86.67% of dual-cycling lipids from DRF mice were phase-inverted, and only 3.33% remained phase-locked compared to NRF (Fig. 6a)". If this the case, any percentage of lipids must be #/10. Additionally, reporting small numbers like this as percentages is a bit hyperbolic, and the exact number should be reported in addition to the percentage.

5. For the analysis of the SA, SD, and KO groups, 40 week old animals were used and a high-fat diet employed for 16 weeks with the rationale (as I understand it) that the HFD increases daytime feeding. These experiments are not rationalized well. Why were middle-age animals used here as compared to the young animals in the earlier studies? Why HFD as opposed to the NRF and DRF groups? Some explanation in the Results as well as implication of the results in the Discussion is also required. Questions about these experiments: Why did the SD group alleviate hyperglycemia compared with the KO? What do the authors make of the 'intermediate' role of the SA? P13, line 436 indicates $P < 0.05$. Was this an adjusted P value? If not, why not? Were all timepoints combined for the statistical test? Why the specific discussion of D-phenylactate when deoxyribonic acid and 4-methylcatechol 2 sulfate are similarly regulated? Finally the authors suggest that this confirms the 'nutrient' sensing role of p-PER2-S971—I'm not sure I follow the reasoning to this conclusion and it should be made more explicit.

Minor comments:

6. P4, line 123: "(no missing values in > 70% of samples)" would be more clearly written as "(<30% missing values)"

7. P5, line 160: "The majority of diurnal phospho-proteins oscillated around ZT4 and ZT8/20" – not clear to me what ZT8/20 means.

8. Figure 6d, it would make more sense to keep the bars colored by lipid group (ie. TAG to have the same color in DRF and NRF etc).

9. Methods for targeted quantification profiling?

10. P14, line 446: 'citate'  "citrate"

11. P22, line 748: This study is not really circadian in the absence of constant conditions, so this is really diurnal oscillation analysis.

REVIEWER COMMENTS

Reviewer #1 (Remarks to the Author):

This paper describes post-translational modifications in the liver shaped by meal timing. The paper is rather a resource than a scientific investigation on a specific question. The only experimental approach on Per2 is rather preliminary.

Re: Firstly, we agree with the reviewer that the paper provides a resource to the community, rather than being a hypothesis-driven study. But time-restricted feeding (TRF) is emerging as an important dietary intervention for the prevention and control of metabolic diseases, as evidenced by the publication of several successful preclinical studies but also by human clinical trials that have reported positive outcomes. So a resource study of a critical organ that may be affected by TRF is an important dataset that is needed by the field. Indeed, so far, proteomics and lipidomics datasets are rare in studying circadian rhythms under TRF. Our work used a proteomics and lipidomics approach to fill this gap in knowledge.

Secondly, correlation across rhythmic omics profiles reveals Per2 Ser971 phosphorylation (Per2-pSer971) as a critical node in the circadian network within the liver among many other signatures. In addition, we generated an antibody recognizing Per2-pSer971 and validated the responsiveness of Per2-pSer971 towards nutrient availability in cultured cells. Thus, it is not clear to us why the reviewer feels this aspect of the report is rather preliminary, especially for a resource study. Indeed, as pointed out by Reviewer 4, *"this is a tremendous effort, and I'm certain the details and connections will be valuable to the scientific community."*

Reviewer #2 (Remarks to the Author):

...The proteomics methodology applied appears standard. However, it'd be helpful to include more details, such as instrument method used for phosphoproteomics experiment.

Response: Thank you for raising these points. All these details have been added to the Methods. Each omics was described in a separate section.

Was it any different from the DDA and DIA used for non-enriched samples?

Response: It is similar to the DDA/DIA but with changes, e.g., the ESI voltage (2.5 kV in DIA proteomics vs 2.3 kV in PTM proteomics) and the number of precursor ions for MS2 analysis. Details are as below.

Phosphoproteomics, DDA

5 mg proteins were aliquoted for trypsin digestion, desalted in the C18 desalting column and lyophilized. Phospho-peptide fractions were enriched in an IMAC-Fe column (Fisher Scientific, #A32992) and incubated at room temperature for 30 min before centrifugation and lyophilization. Shotgun proteomics analyses were performed using an EASY-nLC 1200 UHPLC system (Thermo Fisher) coupled with an Q Exactive HF-X mass spectrometer (Thermo Fisher) operating in the DDA mode via a nano-electrospray ion source (Nanospray Flex, spray voltage of 2.3 kV) at a capillary temperature of 320 °C. 1 µg sample was analyzed by UHPLC at a flow rate of 600 nL/min in a 120 min linear gradient from 5 to 100% of eluent B (0.1% HCOOH in 80% ACN) in eluent A (0.1% HCOOH in H₂O) at a flow rate of 600 nL/min and detailed as follows, 5-10% B, 2 min; 10-30% B, 110 min; 30-50% B, 5 min; 50-95% B, 1 min; 95% B, 5 min. The Q Exactive HF-X was operated in Top30 mode with a full scan of 350-1,500 m/z at a resolution of 120,000 (m/z 200). The AGC target value was set to 3×10⁶ at a maximum ion injection time of 80 ms. Precursor ions were fragmented by HCD at a NCE of 27% and analyzed in MS/MS, with the resolution set to 15,000 (m/z 200), an AGC of 5×10⁴, a maximum ion injection time of 100 ms, an intensity threshold of 5×10³ and a dynamic exclusion parameter of 30 s.

The resulting spectra were searched against the UniProt mouse reference proteome (Proteome ID UP000000589, release 2019.01.18) in Proteome Discoverer (v.2.2). Searched parameters: maximum missed cleavage sites of 2, mass tolerance of 10 ppm for precursor ion scans and a mass tolerance of 0.02 Da for the product ion scans. Cysteine carbamidomethylation (C/+57.021 Da) was set as a fixed modification; N-terminal acetylation (N-terminal/+42.011 Da) as N-Terminal modification; phosphorylation of serine, threonine and tyrosine residue (S, T, Y/+79.966 Da) and methionine oxidation (M/+15.995 Da) as dynamic modifications; FDR < 1% at levels of PSM and protein (matched with at least one unique peptide).

Similarly, how was the data analysis different in Proteome Discoverer for phosphoproteomics, ubiquitylated or succinylated datasets?

Response: The data analysis is highly similar for different PTM datasets. Spectra were searched in Proteome Discoverer against the UniProt mouse reference proteome

(Proteome ID UP000000589) with carbamidomethylation set as a fixed modification; a PTM of interest (p-S/T/Y, GG-K, succinylated-K or deamidated N, details see the next question), N-terminal acetylation and methionine oxidation as variable modifications.

What parameters and search engines were used to detect the various post translational modifications. More details will help the reader understand the data better and it also increases confidence if they try to reproduce the methodology at their end.

Response: Search engines are the same for all proteomics, i.e., Proteome Discoverer. The parameters for PTMs are as follows. The difference is marked as bold characters.

Phosphoproteomics, DDA

Cysteine carbamidomethylation (C/+57.021 Da) was set as a fixed modification; N-terminal acetylation (N-terminal/+42.011 Da) as N-Terminal modification; **phosphorylation of serine, threonine and tyrosine residue (S, T, Y/+79.966 Da)** and methionine oxidation (M/+15.995 Da) as dynamic modifications; FDR < 1% at levels of PSM and protein (matched with at least one unique peptide).

Ubiquitylomics, DDA

Cysteine carbamidomethylation (C/+57.021 Da) was set as a fixed modification; N-terminal acetylation (N-terminal/+42.011 Da) as N-Terminal modification; **GG of lysine (K/+114.043 Da)** and methionine oxidation (M/+15.995 Da) as dynamic modification; FDR < 1% at levels of PSM and protein (matched with at least one unique peptide).

Succinylomics, DDA

Cysteine carbamidomethylation (C/+57.021 Da) was set as a fixed modification; N-terminal acetylation (N-terminal/+42.011 Da) as N-Terminal modification; **succinylation of lysine (K/+100.016 Da)** and methionine oxidation (M/+15.995 Da) as dynamic modification; FDR < 1% at levels of PSM and protein (matched with at least one unique peptide).

N-Glycomics, DDA

Cysteine carbamidomethylation (C/+57.021 Da) was set as a fixed modification; N-terminal acetylation (N-terminal/+42.011 Da) as N-Terminal modification; **glycosylation of asparagine (deamidation ¹⁸O (N)/+2.988 Da)** and methionine oxidation (M/+15.995 Da) as dynamic modification; FDR < 1% at levels of PSM and protein.

Reviewer #3 (Remarks to the Author):

...In general, the analysis "seems to be" sound; however – from my point of view – the manuscript will benefit from certain clarifications.

4.1 Sample exclusion in the analysis:

a) For some data-types, e.g., "Label-free proteomics", samples were removed if their correlation with other samples were below a certain threshold. "N04_1 was removed from phosphoproteomics analysis. N24_4, D16_2, D24_1, and D28_3 were removed from ubiquitylproteomics analysis." How can you be certain the variability observed (related to low correlation) is not biological? A better – beyond a correlation-based – criteria would be required.

Response: Thanks for raising this point. We used the rhythmicity of circadian clock proteins, e.g., PER2, DBP and NR1D1, as a benchmark. Inclusion of these samples did not alter the diurnal rhythmicity of these proteins, as well as the overall distribution and PSEA results; thus, we included all samples for downstream analysis.

b) When conducting the MixOmics analysis: do you consider only samples with all profiles available? How this affects the analysis?

Response: We divided the samples into 3 groups and showed the results from the evening subset of samples based on the balanced error rate of the statistical model. These were clarified in the main text and shown as below.

[Line 302] The modeling applies the *N*-integrative supervised analysis with DIABLO, because the same *N* samples are measured on different 'omics platforms. We divided the input data into three subsets based on the time of day; *i.e.* morning subset (sampled at ZT0 and ZT4, *n* = 16 per group), evening subset (sampled at ZT8 and ZT12, *n* = 16 per group) and night subset (sampled at ZT16 and ZT20, *n* = 16 per group), and fed into the statistical modeling. The results showed that the evening subset returned the best model with an overall balanced error rate (BER) of 0.0312 compared to the other two subsets (morning subset BER = 0.278 and night subset BER = 0.203).

Next, we focused on the mixOmics model based on the evening subset of multi-omics data and found that the statistical model could clearly segregate TRF groups in the phospho-proteome and proteome dataset, and to a lesser degree in the ubiquityl-proteome (**Fig. 4f**). Importance plotting shows the rank and weight of the diurnal proteins that distinguish DRF from NRF. We found that phosphorylated heme-binding protein 1 (p-HEBP1) and phosphorylated Rev-Erb α /NR1D1 (p-NR1D1), which is a nuclear receptor for heme⁴⁹, and p-PER2 contributed the most (**Fig. 4g**).

4.2 Feature selection:

There are several methods mentioned for `MetaCycle::meta2d` function, `RAIN::rain`, and `Circacompare::circacompare`.

a. Please clarify the selection criteria: from the reading, I understand a feature is selected only if it is identified as significant by the three methods.

Response: We have clarified the selection criteria in the Methods, main text and figure legends. Statistical significance is defined by all three methods; *i.e.*, circadian rhythmicity was determined by the algorithms `MetaCycle` (adjusted P -value < 0.05), `RAIN` (adjusted P -value < 0.05) and `CircaCompare` (P -value < 0.05).

b. Data is collected over 48 hours, but then in the Results section is mentioned the analysis for 48 hours, and in the Methods section, it is mentioned a 24-hour based analysis: can you clarify? Overall I would rephrase the question: please clarify when do you use a 24-hour analysis and when 48 hours?

Response: Yes, we used the 24-hour analysis; *i.e.*, period length is defined as 24 h. Proteomics and lipidomics are collected from two different cohorts, the former spanning 44 h (ZT0 to ZT44) and the latter 24 h (ZT0 to ZT24). We have clarified this point in the main text, Methods and Fig. 1a.

Note: Samples collected over 48 hours would provide more robust results in circadian rhythm studies, compared to 24 hours, because it can show the rhythm persists in the next day. Nevertheless, 24 hours is the standard in circadian studies.

4.3 Integration with other publicly available omic features.

While PTM is systematically characterized, and there is mention of transcriptomic analysis (some of them by the same team in the interesting paper of *iScience* 2021), I am missing a data-driven comparison between both omic beyond the mentions in the manuscript.

Something along Protein vs. gene transcripts, or at pathway level, would help to visualize at what level both omics are coordinated or not.

Response: Thanks for this suggestion. We performed these analyses and showed the results of this proteome vs transcriptome comparison including the gene-level and pathway-level comparisons in Fig. 7 and quoted as below.

Integrative Analysis of Diurnal Transcriptome and Multi-level Proteomes

[Line 406] Next, we compared this diurnal multi-level proteomics dataset with a recently published diurnal transcriptomics dataset (GSE150380)¹⁴ and found that rhythmic proteins (including unmodified and one of the four PTMs) and rhythmic mRNAs had 489 matched pairs based on the source gene (Fig. 7a). A comparison among rhythmic unmodified

proteins, rhythmic PTM proteins and rhythmic mRNAs revealed that 77 genes exhibited diurnal rhythms in all three groups and 947 between two groups (**Fig. 7a**). In contrast, 2294 rhythmic mRNAs did not have matched diurnal rhythmic proteins (**Fig. 7a**).

Among the 489 mRNA-protein pairs, there is one gene (the gluconeogenic gene *Pck1*) that exhibited diurnal rhythmicity at four and the most levels of omics, followed by five genes (*Plin5*, *Got1*, *Sun2*, *Hmgcr* and *Abcg5*) with diurnal activity across three levels of omics (**Fig. 7b**). Phase plot showed that almost all these genes exhibited various degrees of phase delay from a transcript-level rhythm to protein-level rhythms (**Fig. 7b**). This is matched with the patterns seen in circadian clock genes, such as *Per2* and *Nr1d1*. Remarkably, the succinylated PCK1 rhythm exhibited an 8-h phase delay compared to its mRNA rhythm under NRF (**Fig. 7b and Fig. 7c**). The glutamic-oxaloacetic transaminase gene *Got1* is an exception in that glycosylated GOT1 (ZT8.3) rhythm was 10.7 h advanced in phase compared to either protein or mRNA rhythms (**Fig. 7b and Fig. 7c**).

We analyzed the mRNA-protein pairs and found that the connections between diurnal multi-level proteomes and transcriptome exhibited a similar pattern between DRF and NRF (**Fig. 7d**). Transcript rhythms were mainly matched with rhythms in the unmodified protein or phosphorylated proteins, whereas unmodified protein rhythms were mainly matched with rhythms in mRNA, phosphorylated or ubiquitylated proteins. These findings revealed the complex network with regards to the biogenesis of diurnal rhythms in the liver.

Next, we compared the rhythmic pathways among different levels of omics, as measured by PSEA of rhythmic proteins and genes (Kuiper test, $q < 0.05$). Like with the gene-level comparison, most pathways had no matched rhythms across the omics under either DRF or NRF (**Fig. 7e**). Connectivity maps for the rhythmic pathways (224 and 185 for DRF and NRF, respectively) revealed that strong connections under NRF including the mRNA-phosphorylation and phosphorylation-ubiquitylation links were replaced by connections such as the unmodified protein-glycosylation and phosphorylation-glycosylation links under DRF (**Fig. 7f**). Comparative analysis of rhythmic pathways between transcriptome and proteomes showed that non-coding RNA processing and metabolic process (ZT8-10) were found at the transcript and phosphorylated protein levels under NRF, while pathways related to translation (ZT18-20) were connected between transcript and protein levels under DRF (**Supplementary Fig. 7a**).

6) Small modifications

Figure 1 Panel a: Please modify the figure. As it is shown, it seems to imply (from my point of view) that NRF is associated with 48h PTM and DRF to the 24h-Lipidomics. I understand it is not, but it will help to modify the panel to clarify that.

Response: Thanks for this suggestion. We have modified the panel in Fig. 1a.

Reviewer #4 (Remarks to the Author):

...While the manuscript is well written, there is a lot of data to assimilate, and other than gross conclusions (such as the over-representation of lipid data in the proteomics experiments) it is difficult to connect the various data types. Nevertheless, this is a tremendous effort, and I'm certain the details and connections will be valuable to the scientific community.

Specific comments for improvement:

1. Given the importance and sexual dimorphism in circadian studies, I applaud the authors study of female mice in their experiments. I was however not able to find complete support for the statement that “female mice exhibit a greater consolidation of circadian rhythms of peripheral clock oscillations during DRF compared to males” on lines 89/90 of p3, with a reference to one of their previous studies (Xin et al, iScience, 2021). The iScience article states that “Together, the liver clock and part of the adipose clock maintained similar responses to inverted feeding between male and female mice; however, peripheral clocks in kidney and heart seemed to be less robust under DRF in males.” It is important to be more precise in describing the premise of this study, given that the liver clock is not strongly impacted by DRF in the authors’ previous work and is the only organ studied here.

Response: Thank you for raising this point. We added more references to support this claim in the manuscript and as below.

[Line 90] Sex differences in the robustness of circadian rhythms have been increasingly recognized. In mammals, circadian rhythms of locomotion, endocrine system, cellular metabolism and gene expression are more sustained in females than males¹⁻³. Notably, female mice exhibit a greater consolidation of diurnal rhythms of non-hepatic peripheral oscillators during DRF compared to males⁴⁻⁶. In the meanwhile, females are much less represented than males in circadian research⁷. Nevertheless, the liver clock and transcriptome exhibit comparable and complete entrainment to DRF from both sexes⁴⁻⁶. Thus, we used livers from female mice as a model to unmask the fine regulation of diurnal rhythms with respect to phase regulation.

References:

1. Weger, B.D., Gobet, C., Yeung, J., Martin, E., Jimenez, S., Betrisey, B., Foata, F., Berger, B., Balvay, A., Foussier, A., et al. (2019). The Mouse Microbiome Is Required for Sex-Specific Diurnal Rhythms of Gene Expression and Metabolism. *Cell Metab.* 29, 362-382.e8. 10.1016/j.cmet.2018.09.023.
2. Anderson, S.T., and FitzGerald, G.A. (2020). Sexual dimorphism in body clocks. *Science* (80-.). 369, 1164–1165. 10.1126/SCIENCE.ABD4964.
3. Talamanca, L., Gobet, C., and Naef, F. (2023). Sex-dimorphic and age-dependent

- organization of 24-hour gene expression rhythms in humans. *Science* (80-.). 379, 478–483. 10.1126/science.add0846.
4. Xin, H., Deng, F., Zhou, M., Huang, R., Ma, X., Tian, H., Tan, Y., Chen, X., Deng, D., Shui, G., et al. (2021). A multi-tissue multi-omics analysis reveals distinct kinetics in entrainment of diurnal transcriptomes by inverted feeding. *iScience* 24, 102335. 10.1016/j.isci.2021.102335.
 5. Manella, G., Sabath, E., Aviram, R., Dandavate, V., Ezagouri, S., Golik, M., Adamovich, Y., and Asher, G. (2021). The liver-clock coordinates rhythmicity of peripheral tissues in response to feeding. *Nat. Metab.* 3, 829–842. 10.1038/s42255-021-00395-7.
 6. Li, M.-D. (2022). Clock-modulated checkpoints in time-restricted eating. *Trends Mol. Med.* 28, 25–35. 10.1016/j.molmed.2021.10.006.
 7. Obodo, D., Outland, E.H., and Hughey, J.J. (2023). Sex Inclusion in Transcriptome Studies of Daily Rhythms. *J. Biol. Rhythms* 38, 3–14. 10.1177/07487304221134160.

2. The mice used in the study appear to be relatively young at 7 weeks. Is there a particular reason to use animals of this age? What is the status of the estrus cycle in animals this age, and could there be any impact on the results?

Response: We used animals of this age because 7 weeks (or postnatal day 50) is considered as sexually mature (Brust et al, *Front Zool*, 2015, PMID: 26816516). Animals would be acclimated to the facility for 7-10 days and subjected to time-restricted feeding for 7 days. At the end of TRF, they are more than 9 weeks old, which is considered adulthood.

The estrus cycle typically begins between postnatal day 25 and 40 with a period of 4-5 days (*Biology of the Laboratory Animals*, 2nd Edition, Chapter 11 Reproduction, <https://www.informatics.jax.org/greenbook/>). Unsynchronized estrus cycle could impact body weight and neuroendocrine system; however, to our knowledge, its impact on the circadian clock has not been adequately studied.

To synchronize the estrus cycle of our animals, we co-housed these mice in a big rodent cage for 5-7 days before relocating to regular cages ($n = 4$ per cage) in our SPF facility. When assigning the mice to different groups, we randomized and controlled the body weight statistics to make sure the body weight is comparable between NRF and DRF groups. We have clarified these measures in the Methods section.

3. The method used to assess rhythmicity is not clearly described, which is an important component of the paper. The authors report using three methods (RAIN, CircaCompare, Metacycle) and that “Adjusted p-values were optimized by inspecting the visual fitness in the exploratory Nitecap platform.” The methods have more specific details on the cutoffs used (which is appreciated, and the fact these are available in the methods should be referenced in the main text with the previous sentence). At the same time, it is a bit disconcerting to see cut-offs based on somewhat arbitrary visual inspection so this needs to be better rationalized.

Response: Thank for this suggestion. We added the *P*-values in the main text as well

as in the figure legends. We applied adjusted *P*-values across the omics datasets for a more rational determination of the rhythmicity. It is specifically described as below.

Circadian rhythmicity was determined by the algorithms MetaCycle (adjusted *P*-value < 0.05), RAIN (adjusted *P*-value < 0.05) and CircaCompare (*P*-value < 0.05).

4. I am a little confused by the definition of dual-cycling. For example, the legend for Figure 6a indicates the phase shift for dual-cycling lipids, which I assume to be a histogram. Counting the number of samples, there appears to be a total of 10 lipids, which is consistent with the heatmap of 6b. However, the percentages reported do not make sense-- P11, lines 370/371: "Strikingly, 86.67% of dual-cycling lipids from DRF mice were phase-inverted, and only 3.33% remained phase-locked compared to NRF (Fig. 6a)". If this the case, any percentage of lipids must be #/10. Additionally, reporting small numbers like this as percentages is a bit hyperbolic, and the exact number should be reported in addition to the percentage.

Response: We clarified this point by rephrasing the figure legend indicating it is a histogram, and reporting the exact number next to total, like x/10. See Fig 2c, 3c, 4b, 5e and Fig. 6a.

5. For the analysis of the SA, SD, and KO groups, 40 week old animals were used and a high-fat diet employed for 16 weeks with the rationale (as I understand it) that the HFD increases daytime feeding. These experiments are not rationalized well. Why were middle-age animals used here as compared to the young animals in the earlier studies? Why HFD as opposed to the NRF and DRF groups? Some explanation in the Results as well as implication of the results in the Discussion is also required.

Response: Thanks for raising these points. We removed the data for the 40-week-old animals.

Finally the authors suggest that this confirms the 'nutrient'-sensing role of p-PER2-S971—I'm not sure I follow the reasoning to this conclusion and it should be made more explicit.

Response: Thanks for raising this point. Cellular experiments in Fig. 7g showed that PER2-pSer971 is activated by glucose treatment in AML-12 hepatocytes. These findings demonstrate that PER2-pSer971 senses nutrient availability.

Minor comments:

6. P4, line 123: "(no missing values in > 70% of samples)" would be more clearly written as "<30% missing values)"

Response: Corrected.

7. P5, line 160: "The majority of diurnal phospho-proteins oscillated around ZT4 and ZT8/20" – not clear to me what ZT8/20 means.

Response: This statement is corrected as follows. *Most diurnal phospho-proteins oscillated around ZT4 under DRF and at ZT8 and ZT20 under NRF (Supplementary Fig. 2a).*

8. Figure 6d, it would make more sense to keep the bars colored by lipid group (i.e. TAG to have the same color in DRF and NRF etc).

Response: Corrected.

9. Methods for targeted quantification profiling?

Response: We have added the methods for targeted quantification profiling.

Targeted profiling of NAE and NAPE

[Line 744] N-acylethanolamines (NAEs) and N-acyl phosphatidylethanolamines (NAPEs) were extracted from liver tissues using a modified method of Bligh and Dyer's extraction⁶⁷. The organic phase was extracted and dried in SpeedVac under organic mode. Samples were resuspended in acetonitrile: isopropanol (1:2 v/v) containing appropriate concentrations of internal standards including d7-PE33:1 from Avanti Polar Lipids, d8-20:4-EA, d4-16:0-EA, d4-22:6-EA and d5-MAG-20:4 from Cayman Chemicals. Samples were analyzed on a ThermoFisher U3000 DGLC coupled to Sciex 6500 Plus QTRAP under the electrospray ionization mode. Individual lipids were separated on an Agilent Zorbax Eclipse Plus column (100 x 2.1 mm, 1.8 µm) using Mobile Phase A (10 mM ammonium formate: acetonitrile: isopropanol 50:30:20, pH 8) and Mobile Phase B (10 mM ammonium formate: acetonitrile: isopropanol 1:9:90, pH 8). NAEs and NAPEs were quantitated by referencing to spiked internal standards.

10. P14, line 446: 'citate'  "citrate"

Response: The HFD data were removed.

11. P22, line 748: This study is not really circadian in the absence of constant conditions, so this is really diurnal oscillation analysis.

Response: Thanks for this suggestion. We changed circadian to diurnal in the manuscript.

REVIEWERS' COMMENTS

Reviewer #3 (Remarks to the Author):

All my previous comments have been addressed.

I can also observe that comments from fellow reviewers were also addressed.

Also, reading the updates based on Reviewer 1 and 4, it comes clear the added value of the final experiments.

Reviewer #4 (Remarks to the Author):

The authors have adequately addressed the previous concerns.